# Broadly conserved protective epitopes on the lyme disease vaccine antigen, OspA

**Graham G. Willsey[1], Michael J. Rudolph[2], Carol Lyn Piazza[1], Yang Chen[2], Grace Freeman-Gallant[1], Lisa A. Cavacini[3], David J. Vance[1], Nicholas J. Mantis[1]\***

**1** Division of Infectious Diseases, Wadsworth Center, New York State Department of Health, Albany, New York, United States of America, **2** New York Structural Biology Center, New York, New York, United States of America, **3** Department of Medicine, University of Massachusetts Chan Medical School, Worcester, Massachusetts, United States of America

\* nicholas.mantis@health.ny.gov

## Abstract

Lyme disease, caused by the spirochete, *Borrelia burgdorferi* sensu latu (Bbsl), is a tickborne infection of increasing incidence in North America, Europe and Asia. While vaccines based on Outer surface protein A (OspA) have proven highly efficacious at blocking Bbsl tick-to-human transmission, the high degree of antigenic variability among the major OspA serotypes (ST) has made the development of a broadly cross protective vaccine difficult. Recent profiling of protective human monoclonal antibodies (mAbs) has suggested the existence of conserved epitopes situated within OspA's central β-sheet (CBS), although direct comparisons of cross-serotype functionality has been hindered by biological differences among the major Bbsl genospecies. To address these issues, we developed a panel of isogenic *B. burgdorferi* reporter strains expressing the seven major OspA serotypes (ST1–7) and probed them with CBS-targeting mAbs to evaluate their complement-dependent borreliacidal activity. The mAbs segregated into three distinct classes: class 1 mAbs exhibited potent killing against all seven OspA serotypes, while classes 2 and 3 had restricted or no activity against two of the seven serotypes. Structural analysis of Fabs derived from each class of mAbs in complex with OspA ST1 showed that they target overlapping epitopes spanning β-strands 6–10 and involve contact with largely invariant residues. Further analysis of *B. burgdorferi* reporter strains expressing OspA variants from 17 additional Bbsl genospecies identified Lys-107 as a determinant of susceptibility for nearly all CBS mAbs. Taken together, these findings raise the prospect of structure-based design of a broadly protective monovalent Lyme disease vaccine.

## Author summary

Lyme disease, caused by the spirochete *Borrelia burgdorferi* sensu latu (Bbsl), is a tickborne infection of increasing incidence in North America, Europe and Asia.

**Data availability statement:** All data underlying the study are provided in the manuscript and supplementary information. The protein structures generated in this study were deposited in the Protein Data Bank (PDB; http://www.rcsb.org/pdb/) under accession numbers 8TUZ, 8TV3, 8TVD, and 8TVJ.

**Funding:** This work was supported by the National Institute of Allergy and Infectious Diseases (NIAID) National Institutes of Health (NIH), Department of Health and Human Services (Contract No. 75N93019C00040 to NJM). This content is solely the responsibility of the authors and does not necessarily represent the official views of the NIH. The funders had no role in study design, data collection and analysis, decision to publish, or preparation of the manuscript.

**Competing interests:** The authors have declared that no competing interests exist.

While vaccines based on Outer surface protein A (OspA) have proven highly efficacious at blocking Bbsl tick-to-human transmission, the genetic and serological heterogeneity of OspA across *Borrelia* genospecies has complicated matters. In this report, we use a collection of transmission-blocking human monoclonal antibodies to delineate a protective region (epitope) within the central core of OspA that is conserved across all major OspA serotypes. These results have important implications for engineering a broadly reactive Lyme disease vaccine.

## Introduction

The spirochete *Borrelia burgdorferi* sensu stricto (Bbss) is the primary etiologic agent of Lyme disease, the most common tick-borne infection in the United States [1]. The disease typically presents with a characteristic "bullseye" rash (erythema migrans) before potentially progressing to a disseminated infection, which may result in carditis, neuroborreliosis, and/or Lyme arthritis [2]. Although antibiotic treatment is generally effective at any stage of infection, a subset of individuals experience persistent symptoms known as post-treatment Lyme disease (PTLD) syndrome [3]. In Europe and Asia, Lyme disease is also caused by three additional genospecies within the *Borrelia burgdorferi* sensu lato (Bbsl) complex: *B. afzelii*, *B. garinii*, and *B. bavariensis*. While these four genospecies share key pathogenic traits, they are genetically and antigenically heterogeneous and exhibit distinct clinical manifestations. This heterogeneity has posed significant challenges to the development of broadly protective Lyme disease vaccines suitable for use across North America, Europe, and Asia.

The only licensed Lyme disease vaccine for human use in the United States was LYMERix, which was based on outer surface protein A (OspA) serotype 1 (ST1), the predominant OspA ST expressed by *B. burgdorferi* in North America [4,5]. OspA is a ~31 kDa lipoprotein abundantly expressed on the surface of *B. burgdorferi* in the midgut of unfed ticks. During feeding, OspA expression is downregulated as the spirochete migrates through the hemolymph and salivary glands. Nonetheless, transmission is effectively blocked if a tick feeds on a host with pre-existing OspA antibodies induced by vaccination [6–10]. Similarly, administration of certain mouse or human monoclonal antibodies (mAbs) targeting OspA prior to tick challenge is sufficient to protect rodents and non-human primates from *B. burgdorferi* infection [7,11,12]. While the precise mechanisms by which these antibodies block transmission are not fully understood, evidence supports a model in which they engage spirochetes within the tick midgut [10,13].

As the incidence of Lyme disease continues to rise in both the United States and Europe, there is renewed interest in developing OspA-based vaccines. To provide broad protection beyond North America, such vaccines must target not only *B. burgdorferi* (ST1), but also six additional OspA serotypes (ST2–7) associated with *B. afzelii* (ST2), *B. garinii* (ST3, ST5–7), and *B. bavariensis* (ST4) [14,15]. Research dating back over three decades has shown that OspA-based immunization typically confers strong protection against homologous serotype challenge but limited or no

protection against heterologous serotypes [16–18]. These observations, combined with the identification of immunodominant, highly protective B cell epitopes localized to OspA's C-terminal region, have driven the design and evaluation of multivalent OspA vaccines [9,19–22]. One such candidate, VLA15, which incorporates chimeric heterodimers of the C-terminal third of OspA ST1–6, is currently in Phase 3 clinical trials [23–26].

Monoclonal antibodies (mAbs) have been instrumental in identifying cross-protective epitopes on highly variable surface antigens from pathogens such as influenza virus hemagglutinin [27,28], HIV-1 envelope glycoprotein [29], and *Plasmodium* species [30], among others [31]. In the case of Lyme disease, we employed a collection of human mAbs alongside alpaca-derived single-domain antibodies ($V_H$Hs) generated against OspA serotype 1 (OspA$_{ST1}$) to derive the first detailed epitope map of OspA$_{ST1}$ [11,12,32–34]. Three spatially distinct epitope "bins" encompassing more than half the predicted surface-exposed area of OspA were identified [32]. Bin 1 corresponds to OspA$_{ST1}$'s central β-sheet (β-strands 8–10), Bin 2 corresponds to β-strands 11–13, and Bin 3 to β-strands 16–20 and the C-terminal α-helix. To identify possible epitopes shared across serotypes, Wang and colleagues screened the collection of human mAbs for complement-dependent borreliacidal activity against Bbsl strains expressing different OspA serotypes: *B. burgdorferi* (ST1), *B. afzelii* (ST2), and *B. bavariensis* (ST4) [11]. Monoclonal antibodies from Bin 1 (e.g., 221-7, 857-2) had potent borreliacidal activities across all three Bbsl strains, whereas mAbs targeting bins 2 and 3 (e.g., LA-2) were only active against *B. burgdorferi* (ST1), demonstrating that certain epitopes within Bin 1 are conserved across OspA serotypes. In support of this notion, we demonstrated by ELISA using recombinant OspA antigens that 857-2 reacts with all seven OspA serotypes [34]. The structural elucidation of 221-7 Fab-OspA$_{ST1}$ co-complex not only underscored the possibility that there are epitopes within Bin 1 that are conserved across OspA but that these epitope(s) may also be broadly protective [12]. 221-7, for example, blocks tick-mediated transmission of *B. burgdorferi* (ST1) in both mouse and a non-human primate models of pre-exposure prophylaxis [11,12]. In this report we sought to better define the Bin 1 mAbs and their potential to impart complement-dependent borreliacidal activities against the seven major OspA serotypes.

## Results

Differences in *Bbsl* genospecies growth rates, intrinsic serum sensitivities, and susceptibility to commercially available complement sources have hindered the development of a standardized OspA binding and complement-dependent bactericidal assay suitable for all Bbsl serotypes and in silico types (ISTs) [14,15,23,25,35]. To resolve this issue, we constructed a panel of isogenic *B. burgdorferi* reporter strains expressing each of the seven major OspA serotypes (OspA$_{ST1–7}$). This was accomplished by modifying an existing IPTG-inducible *mScarlet-I* viability reporter plasmid (pGW189) to encode different *ospA* alleles under the control of the native *ospAB* promoter from *B. burgdorferi* B31 [34]. Reporter plasmids encoding *ospA* alleles from *B. burgdorferi* B31 (ST1), *B. afzelii* PKo (ST2), *B. garinii* PBr (ST3), *B. bavariensis* PBi (ST4), *B. garinii* PHei (ST5), *B. garinii* DK29 (ST6), and *B. garinii* T25 (ST7) were introduced into *B. burgdorferi* strain HB19-R1, a high passage North American isolate that lacks *ospA* due to the loss of lp54 (**S1** and **S2 Tables**) [36,37].

To validate OspA expression on the surface of the reporter strains, we assessed the reactivity of two mAbs, 857-2 and LA-2, against the panel and compared their profiles to corresponding native Bbsl strains (Fig 1 and S1 Fig). 857-2 was reactive with all seven OspA serotypes expressed in the HB19-R1 background, with median fluorescence intensity (MFI) values ranging from 2500-6400. In contrast, LA-2 was only reactive with OspA$_{ST1}$ (>70% reactivity; MFI > 7,000). Neither 857-2 nor LA-2 reacted with HB19-R1 carrying the empty vector (MFI < 55). The same OspA reactivity profiles were observed when 857-2 and LA-2 were used to probe the native Bbsl strains (**S1 Fig**): 857-2 recognized all seven OspA serotypes, while LA-2 only recognized OspA$_{ST1}$. These results not only demonstrate that the HB19-R1 reporter strains are reliable proxies for assessing antibody binding to different OspA serotypes but also confirm that 857-2 can recognize the seven major OspA serotypes [34].

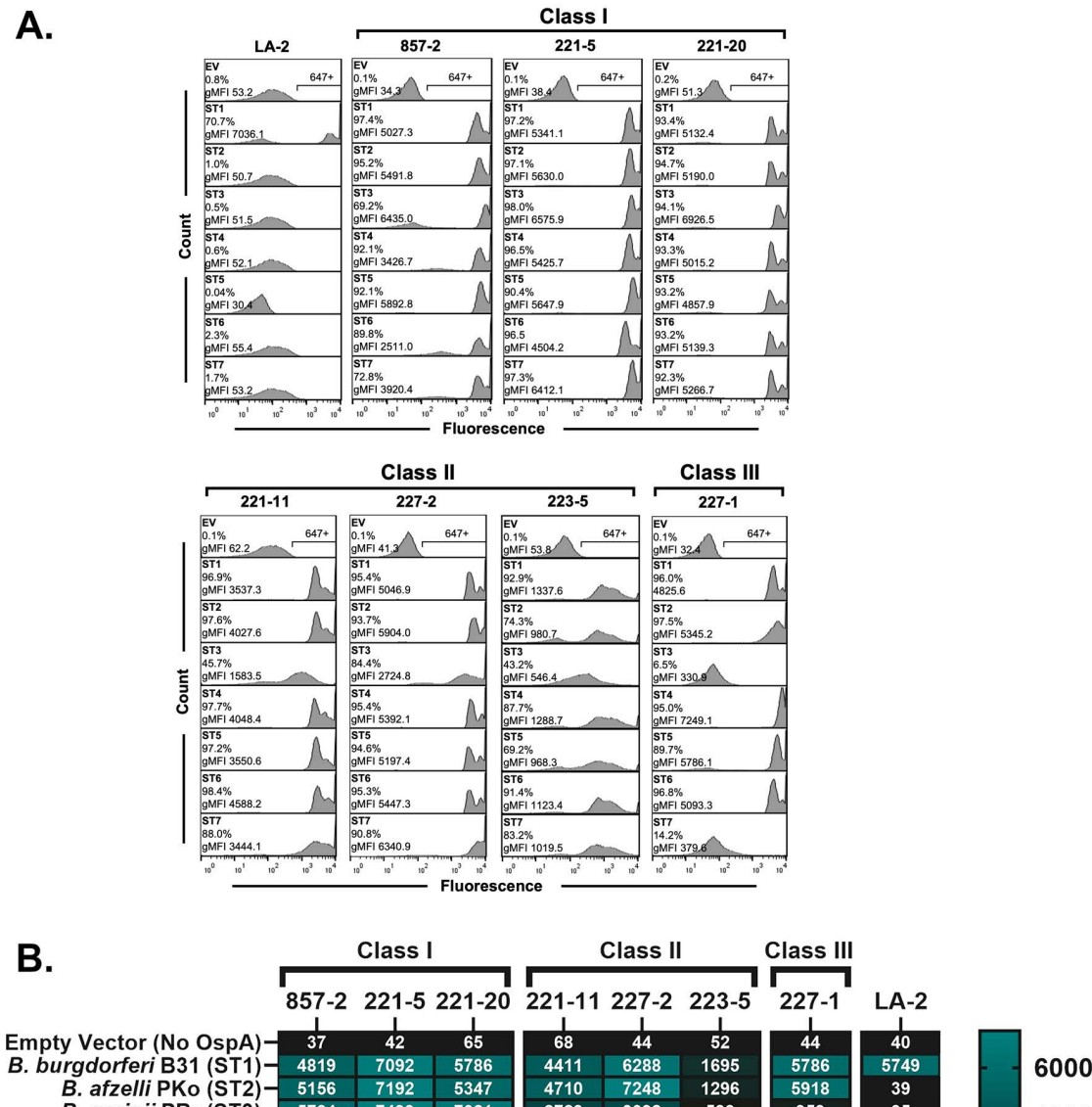

**Fig 1. Surface binding profiles of anti-OspA$_{ST1}$ Bin 1 mAbs to *B. burgdorferi* OspA ST1-7. (A)** Representative flow cytometric histograms of live *B. burgdorferi* HB19-R1 reporter strains expressing OspA ST 1-7, each probed with 10 μg/ml of mAbs indicated in bold text above each column, followed by Alexa 647-labeled goat anti-human IgG secondary antibody. HB19-R1 harboring the empty vector (EV) was used as a negative control. The horizontal bracket represents the region of events positive for fluorescence labeling (647+) on the subsequent plots. The percent (%) of events positive for Alexa 647 fluorescence labeling and the geometric mean fluorescence intensity (gMFI) are shown in the top left corner of each box. Compared to the empty vector (EV) controls, the gMFI values were significantly greater for binding of LA-2 to ST1 (p<0.0001); 857-2 to ST1-7 (p<0.0001); 221-5 to ST1-7 (p<0.0001); 221-20 to ST1-7 (p<0.0001); 221-11 to ST1 (p<0.001), ST2, 4-7 (p<0.0001), and ST3 (p<0.05); 227-2 to ST1-2, 4-7 (p<0.0001) and ST3 (p<0.05); 227-1 to ST1-2, 4-6 (p≤0.0001). N=3. Statistical comparisons were performed using two-way ANOVA with Dunnet's multiple-comparison test. Panels and plots were assembled using FlowJo. **(B)** Heat map summarizing the average gMFIs (with scale shown on right) of *B. burgdorferi* HB19-R1 reporter strains probed with each of the different mAbs depicted in Panel **A.** The results per box are the average of three biological replicates.

The HB19-R1 reporter strains were then probed with the additional mAbs targeting OspA$_{ST1}$'s central β-sheet (so-called Bin 1 mAbs) to assess breadth of OspA serotype recognition (**Table 1**). Interestingly, the mAbs fell into three distinct classes (**Fig 1**). Class I mAbs (857-2, 221-5, 221-20) reacted strongly with OspA$_{ST1-7}$ with MFIs > 4000, while the Class II mAbs (221-11, 227-2, 223-5) had diminished recognition of OspA$_{ST3}$. A single mAb, 227-1, was defined as Class III because of significantly reduced binding to OspA$_{ST3}$ and OspA$_{ST7}$. To determine if these class designations are a function of mAb binding to the different OspA serotypes and not an artefact of expressing heterologous OspA types on the surface of HB19-R1, we evaluated mAb binding to recombinant OspA$_{ST1-7}$ by microsphere immunoassays (MIA). The mAb binding profiles to recombinant OspA$_{ST1-7}$ mirrored exactly those observed with the HB19-R1 reporter strains (**S2 Fig**). These results indicate that a subset of Bin 1 mAbs, namely those designated as Class I (857-2, 221-5 and 221-20), recognize epitopes that are conserved across the major OspA serotypes.

## Cross-serotype borreliacidal activity associated with Bin 1 mAbs

We next wished to determine the functional capacity of the Bin 1 mAbs to promote complement-dependent borreliacidal activity across the different OspA serotypes. To validate the HB19-R1 OspA reporter strains for this purpose, we compared complement-dependent killing of an infectious *B. burgdorferi* B31-5A4-based reporter strain [34] to the HB19-R1 OspA$_{ST1}$ derivative. The assays were performed under conditions optimized for each strain, due to slight differences in growth kinetics and intrinsic complement sensitivities. Cultures were mixed with serial mAb dilutions (0.08-10 nM) in medium containing 10% (B31-5A4) or 2.5% (HB19-R1) guinea pig complement. The following day, IPTG was added to the cultures to induce *mScarlet-I* expression and MFI was measured 24 h (HB19-R1) or 48 h (B31-5A4) later, as described in the Materials and Methods. Under these conditions, the five Bin 1 mAbs had EC$_{50}$ values ranging from 0.63 to 1.25 nM against both the B31-5A4 and HB19-R1 OspA$_{ST1}$ reporter, while an IgG isotype control (PB10) had no effect on spirochete viability (**Fig 2A and 2B**). Moreover, the EC$_{50}$ values for 857-2 and 221-7 obtained using the HB19-R1 OspA$_{ST1}$ strain closely match those reported for *B. burgdorferi* B31 derived using a completely different viability assay [11]. Collectively, these results demonstrate that OspA expression is comparable in both *B. burgdorferi* genetic backgrounds. Thus, the HB19-R1 reporter strains served as reliable readouts of OspA-mediated complement-dependent borreliacidal activity.

We therefore used the panel of HB19-R1 OspA reporters to assess the ability of the full collection Bin 1 mAbs (8 in total) to confer cross-serotype, complement-dependent borreliacidal activity (**Fig 2C**). The results reflected the OspA binding profiles in that **Class I** mAbs (857-2, 221-20, and 221-5) exhibited potent borreliacidal activity against all seven

**Table 1. Characteristics of OspA mAbs used in this study.**

| mAb | bin[a] | class[b] | $K_D$[c] | iGL[d] V$_H$ | iGL V$_L$ | HCDR3 | PDB |
|---|---|---|---|---|---|---|---|
| 221-7[e] | 1 | (2) | 0.66 | HV5–51 | KV1–13 | ARGILRYFDWFL<u>D</u>Y | 7JWG |
| 857-2 | 1 | 1 | 0.30 | HV5–51 | KV1–13 | ARHITTHTYRGFF<u>D</u>F | 8TUZ |
| 221-5 | 1 | 1 | 0.52 | HV5–51 | KV3–11 | ARHLGIDYFFGM<u>D</u>V | 8TV3 |
| 221-20 | 1 | 1 | 0.37 | HV5–51 | KV3–11 | ARHLGIDYFFGL<u>D</u>V | n.a. |
| 221-11 | 1 | 2 | 0.35 | HV5–51 | KV1–13 | VRGIGRYSNWFL<u>D</u>Y | 8TVD |
| 227-2 | 1 | 2 | 0.48 | HV5–51 | KV1–13 | VRSISRSGPGYILSWFF<u>D</u>Y | n.a. |
| 223-5 | 1 | 2 | 0.17 | HV5–51 | KV1–13 | ARYVTVGAF<u>D</u>F | n.a. |
| 227-1 | 1 | 3 | 1.7 | HV5–51 | KV1–13 | TRSISRSGPGFSLSWFF<u>D</u>Y | 8TVJ |
| LA-2 | 3 | n.a. | 64 | n.a. | n.a. | n.a. | 1FJ1 |

[a], from Haque et al 2022; [b], n.a. not applicable, n.d. not determined; [c], nM; [d], inferred germline sequence; [e], from Schiller et al, 2022; 221-7 inferred to be in class 2, based on S2 Fig;

## A.

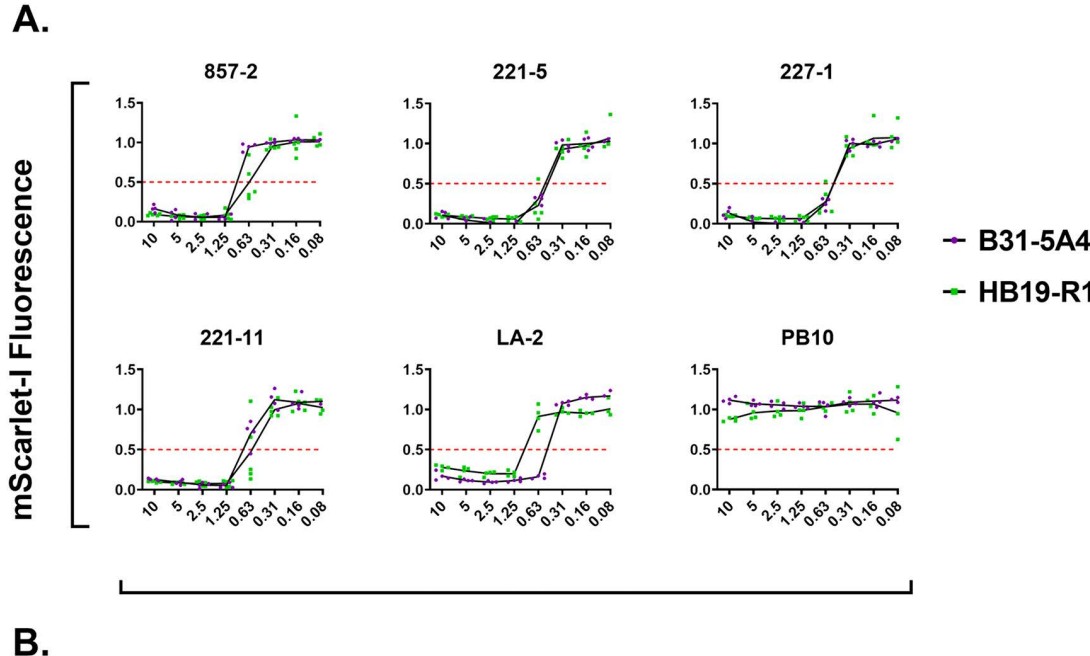

## B.

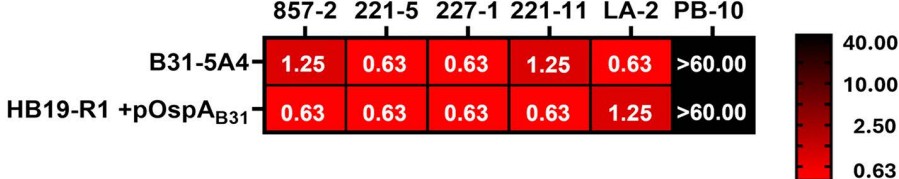

## C.

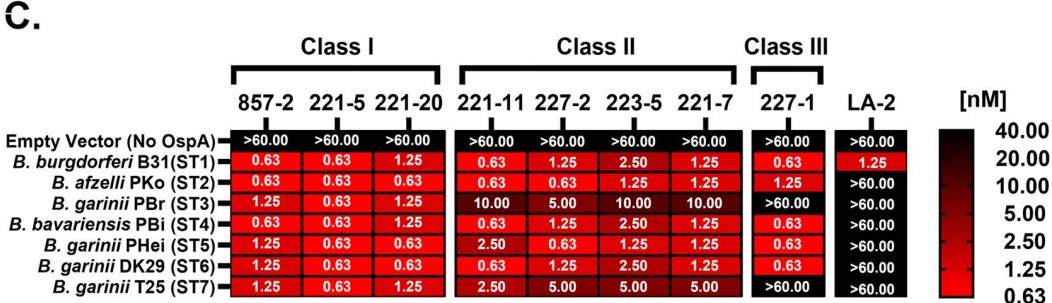

**Fig 2. OspA Bin1 mAbs promote complement-dependent killing of recombinant *B. burgdorferi* strains expressing OspA ST1-7. (A)** Borrelia-cidal mAb titration curves and **(B)** heat map comparing the susceptibilities of an infectious *B. burgdorferi* B31-5A4 *mScarlet-I* viability reporter strain and an HB19-R1 reporter strain expressing the same $OspA_{ST1}$ (B31) allele to complement-dependent killing mediated by $OspA_{ST1}$ mAbs or an isotype IgG1 (PB10) control. Experiments were performed under conditions optimized for each strain as described within the Materials and Methods. $Ec_{50}$ values represent the mean minimum concentration (nM) of mAb resulting in >50% reduction in MFI relative to controls, as determined from 3-5 independent experiments per strain. Statistical differences in mAb susceptibility were assessed via two-way ANOVA followed by Tukey's multiple comparison test. No significant differences were observed. **(C)** Heat map depicting the mean $Ec_{50}$ (nM) of Bin 1 mAbs against recombinant *B. burgdorferi* strains express-ing OspA ST 1-7. Complement-dependent borreliacidal assays were performed with anti-$OspA_{ST1}$ Bin 1 mAbs and a panel of *B. burgdorferi* HB19-R1 (*ospA* negative) strains harboring an IPTG-inducible *mScarlet-I* viability reporter plasmid and expressing *ospA* ST1-7 under conditions described in the Materials and Methods. Controls included an HB19-R1 strain carrying the IPTG-inducible viability reporter plasmid alone (empty vector, *ospA* negative) and a mAb with borreliacidal activity restricted to $OspA_{ST1}$ (LA-2). $EC_{50}$ values shown represent the mean lowest concentration of antibody (nM) resulting in >50% reduction in MFI relative to controls. Reporter strains that exhibited resistance to complement-mediated killing by mAbs at 10 nM were later retested at a single higher dose (66.6 nM). The data shown encompasses 3-5 independent experiments per strain with data normalized as described within the Materials and Methods section. Corresponding antibody titration curves and statistical comparisons are shown in **S3 Fig**.

OspA serotypes, with $EC_{50}$ values ranging from 0.63 to 1.25 nM. Providing further validation of our HB19-R1 reporter system, 857-2's $EC_{50}$ against the ST4 reporter strain is consistent with the previously reported value for *B. bavariensis* strain (PBi), from which the $OspA_{ST4}$ sequence was derived [11]. **Class II** mAbs (221-11, 223-5, 227-2, and 221-7) had reduced potency against OspA ST3 and ST7 ($EC_{50}$ 2.5-10 nM), while 227-1 (**Class III**) was completely devoid of ST3 and ST7 killing.

## Structural basis of $OspA_{ST1}$ recognition by Bin 1 mAbs

To elucidate the epitopes recognized by the three different classes of Bin 1 mAbs, we solved the X-ray crystal structures of Fabs 857-2 (**Class I**), 221-5 (**Class I**), 221-11 (**Class II**), and 227-1 (**Class III**) in 1:1 stoichiometric complexes with $OspA_{ST1}$ at resolutions ranging from 2.2 Å to 3.2 Å (**Tables 2** and **S3** and **Fig 3**). Attempts to solve the X-ray crystal structure of 221-20 Fabs in complex with OspA were unsuccessful. After molecular replacement, the resulting phase information was used to calculate electron density maps employed to manually insert the corresponding residues into each model while manually building additional regions within each Fab-OspA model. Refinement of the four Fab-OspA structures revealed molecular models that possessed excellent geometry and that were consistent with the crystallographic data (**S4 Table**).

In each of the four complexes, the Fabs assumed a canonical structure with two heavy chain immunoglobulin domains ($V_H$, $C_H1$) and two light immunoglobulin domains ($V_L$, $C_L$) each containing 7–9 β-strands arranged in two β-sheets that folded into a two-layer sandwich with all six CDRs (L1-3; H1-3) on one face of each molecule. OspA was comprised of one antiparallel β-sheet with 21 β-strands (referred to as β-strands 1–21) connecting globular N- and C-terminal domains with a single α-helix (referred to as α-helix A) at the C-terminus (**Fig 3**). All 21 β-strands were present in their respective electron density maps of OspA complexed with Fabs 227-1 and 857-2. In the OspA-221-11 complex β-strands 1 and 2 were not visible in the electron density maps, while β-strands 17–21 were absent in the electron density maps of the OspA-221-5 complex, likely because of greater flexibility within these regions of OspA when bound to the particular Fabs. The OspA molecules from each of the four complexes were structurally similar to OspA alone (PDB ID: 2G8C), as evidenced by Root Mean Square Deviation (RMSD) of 0.8 Å to 2.4 Å upon superposition (**S3 Fig**). Thus, the Fabs did not induce any major conformational changes within OspA.

The OspA-Fab structures revealed that 857-2 (**Class I**), 221-5 (**Class I**), 221-11 (**Class II**), and 227-1 (**Class III**) recognize overlapping but distinct epitopes on OspA with similar modes of engagement (**Fig 3**). The four Fabs contact OspA β-strands 6–10 and loop regions between β-strands 7–8 (loop 7–8) and 9–10 (loop 9–10), including residues 87, 100, 102, 105, 106, 107–109, 112, 124, 126, 128–132 (**Fig 3**). All but 857-2 Fab contacts OspA loop 11–12.

Fab 857-2 (**Class I**; $K_D$ 0.30 nM) and 221-5 (**class 1**; $K_D$ 0.52 nM) buried 1,397 Å² and 1,423 Å², respectively, involving CDRs H1, H2, H3, and L2. 857-2 formed seven hydrogen bonds and three salt bridges with OspA, while 221-5 forms 8 hydrogen bonds and five salt bridges with OspA (**Table 2**). The Fabs each form two salt bridges with OspA residue Lys-107; one via CDR-L2 residue Asp/Glu-55 and the other with CDR-H3 residue Asp-109/110 (**Fig 4A-4B**). Fab 221-5 makes

**Table 2. Summary of OspA-Fab interactions.**

| mAb | class [a] | BSA[b] | H-b[c] | SB[d] | SC[e] | PDB ID[f] |
|---|---|---|---|---|---|---|
| 857-2 | 1 | 1397 | 7 | 3 | 0.61 | 8TUZ |
| 221-5 | 1 | 1423 | 8 | 5 | 0.74 | 8TV3 |
| 221-11 | 2 | 1348 | 6 | 3 | 0.52 | 8TVD |
| 227-1 | 3 | 1704 | 5 | 3 | 0.47 | 8TVJ |

[a], Buried surface area (Å²); [b], hydrogen bonds; [c], salt bridges; [d], shape complementarity; [e], Protein Data Bank ID (https://www.rcsb.org/).

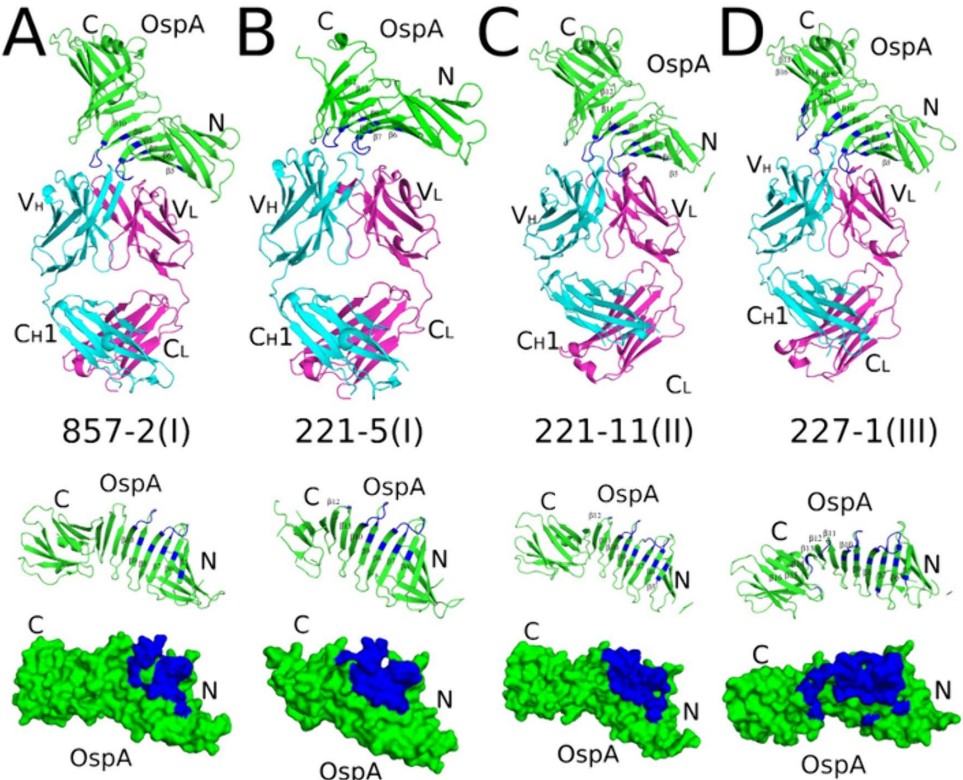

**Fig 3. Crystral structures of Fab-OspA complexes reveal conserved epitopes on OspA$_{ST1}$.** (**top row**) Ribbon diagrams of OspA$_{ST1}$ (green) in complex with Fabs from (A) 857-2 (class **I**), (B) 221-5 (class **I**), (C) 221-11 (class **II**), and (D) 227-1 (class **III**). Fab heavy chains (V$_H$ and C$_H$1) are colored cyan; light chains (V$_L$ and C$_L$) are colored magenta. The OspA N- and C-termini are labelled accordingly, along with selected strand numbers. (**middle row**) The four OspA (green) ribbon representations aligned in the same orientation with respective Fab contract points colored in blue. (**bottom row**) The ribbon and surface representations of OspA highlight the Fab-interacting residues, shown in blue. The N and C-termini of OspA are labelled N and C, respectively. PDB ID for each structure are provided in Table 2.

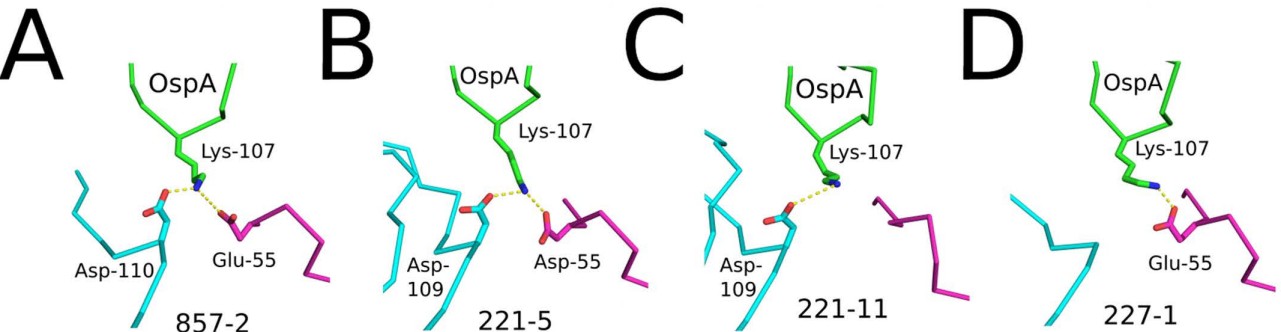

**Fig 4. Key interactions between each Fab and OspA.** Close-up of notable salt bridges between OspA Lys-107 (green) and Fab residues are shown as yellow dashed lines. Fab heavy chains (V$_H$/C$_H$1) are colored cyan, and light chains (V$_L$/C$_L$) are colored magenta. (**A**) Fab 857-2 engages Lys-107 through Glu-55 and Asp-110. (**B**) Fab 221-5 interacts with Lys-107 via Asp-55 and Asp-109. (**C**) Fab 221-11 contacts Lys-107 through Asp-109. (**D**) Fab 227-1 engages Lys-107 through Glu-55. All side chains are drawn as sticks and color coordinated to the main chain color with nitrogen atoms blue and oxygen atoms red.

contacts with Ser-152 in loop 11–12 via framework region (FR) residue Arg-59 but is too distant (4.2 Å) to form an H-bond. 857-2 does not interact with loop 11–12 at all, as Arg-59 is > 9 Å from OspA Ser-152.

Fab 221-11 (**Class II;** $K_D$ 0.35 nM) engages with OspA via CDRs H1, H2, H3, and L2 and has a total buried surface area of 1,348 Å² (**Table 2**). It forms 9 polar contacts with OspA, including a hydrogen bond between CDR-H1 Tyr-32 and OspA Asp-105. Like 857-2 and 221-5, 221-11 also forms a salt bridge between CDR-H3 residue Asp-109 and OspA residue Lys-107 (**Fig 4C**). However, unlike the other two mAbs, 221-11 light chain FR Glu-55 does not form a second H-bond with Lys-107 (**Fig 4C**).

Fab 227-1 (**Class III;** $K_D$ 1.7 nM) stands out from the other three Fabs in that it forms the most extensive interface with OspA, burying a total of 1,704 Å² with contributions from five of the six CDRs (H1-3, L1-2). Fab 227-1 establishes five hydrogen bonds and three salt bridges with OspA, including with Lys-107 (**Fig 4D**). FR residue Arg-59 forms a hydrogen bond with Ser-152 of OspA, as will be discussed in detail below. 227-1 makes additional contacts with OspA not observed in the other three Fabs, including with loop 11–12 (residues 153–154), β-strand 13, loop 13–14 (residues 171–173), and loop 15–16 (residue 193). These additional interfaces result in 239 Å² of total buried surface area. This larger binding surface likely comes at a cost, considering that 227-1 (**Class III**) has a restricted OspA serotype profile (i.e., fails to bind $OspA_{ST3}$ and $OspA_{ST7}$) relative to 857-2 and 221-5, which bind the seven major OspA types.

### Structural basis of cross serotype restriction of Bin 1 mAbs

Based on the structural analysis of the OspA-Fab interactions, we postulate that two distinct molecular interactions account for the reduced binding of 221-11 (**Class II**) and 227-1 (**Class III**) to OspA ST3 and ST7. The first interaction involves the $Glu_{ST1} > Lys_{ST3/7}$ polymorphism at position 131 (**Fig 5A and 5B**). For example, 227-1 $V_H$ FR3 residue Arg-59 forms a salt bridge with Glu-131 in $OspA_{ST1}$, which would be disrupted by the substitution of a Lys at that position. Moreover, a mutation to Lys at position 131 would introduce electrostatic repulsion with Lys-129, which is located ~5 Å away. As a result, the salt bridge between OspA Lys-129 and 227-1 $V_H$ residue Asp-55 would be perturbed. Thus, a single $Glu_{131}Lys$ polymorphism would result in the loss of two salt bridges in the case of 221-11 and 227-1. Second, the $Ser_{ST1} > Asn_{ST3/7}$ polymorphism at position 152 may restrict 221-11 (**Class II**) and 227-1 (**Class III**) recognition of OspA ST3 and 7. 221-11 and 227-1 each form H-bonds with $OspA_{ST1}$ residue Ser-152 via $V_H$ residue Arg-59 (**Fig 5C and 5D**). The Asn polymorphism at this position would perturb a corresponding interaction in $OspA_{ST3}$ and $OspA_{ST7}$.

### OspA recognition is driven by $V_H$ and $V_L$ germline-encoded residues

All eight of the Bin 1 mAbs, including the four whose structures were solved in complex with OspA (857-2, 221-5, 221-11, 227-1) as well as the previously reported 221-7 (PDB 7JWG), are inferred to utilize the same HV5–51 germline lineage (**Table 1** and **S6 Fig**). In the case of 221-5 (**Class I**), the $V_H$ is unmutated from the HV5–51 germline, while the $V_H$ elements of the other four mAbs with solved co-complexes have only nominal mutations (4–6 amino acids) that are not associated with direct OspA interactions. Additionally, the H-CDR3 residue involved in the well-conserved salt bridge with Lys-107, namely Asp109 (or Asp110 or 114, depending on CDR3 length), is encoded by the $J_H$ gene segment in five of the six human germline J genes ($J_H$1 being the exception) (**Table 1**). Similarly, in six of the mAbs, the $V_L$ is derived from the KV1–13 with germline residues involved in critical contacts, including Glu-55 that forms a salt bridge with OspA Lys-107 (**Fig 4**). In the case of 221-5 and 221-20 (**Class I**), which are derived from KV3–11, a single mutation in codon 55 (C > A, resulting in amino acid mutation Ala > Asp) would be sufficient to form a salt bridge with OspA Lys-107. Therefore, BCR combinations with HV5–51 and either KV1–13 or KV3–11 would be expected to have broad affinity for OspA serotypes.

### Borreliacidal epitopes are conserved across OspA in diverse *Bbsl* genospecies

With a detailed structural understanding of the interactions of Bin 1 mAbs with $OspA_{ST1}$, we sought to examine epitope conservation across the *Bbsl* genospecies complex. To accomplish this, we surveyed OspA protein sequence diversity

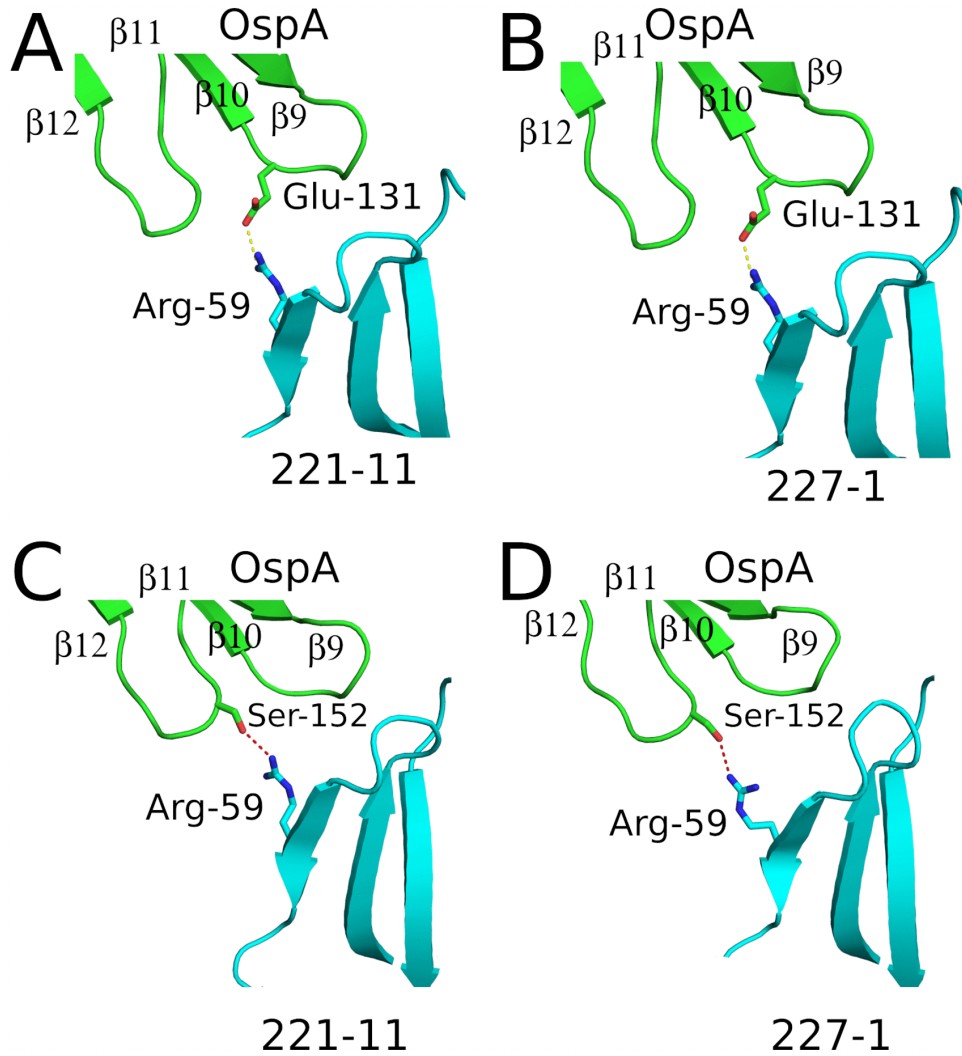

**Fig 5. Influence of OspA primary sequence deviations on antibody function. (A)** Structural basis for Fabs 221-11 and 227-1 cross-reactivity loss in OspA serotypes 3 and 7. Distinct hydrogen bond networks between Fab Arg-59 (cyan) and OspA residues (green) contribute to serotype specificity. **(A)** In Fab 221-11, Arg-59 forms a salt bridge with OspA Glu-131. **(B)** The same Glu-131 interaction is observed in Fab 227-1. **(C)** In 221-11, Arg-59 hydrogen bonds with OspA Ser-152. **(D)** Similarly, Fab 227-1 also engages Ser-152 through Arg-59. The primary sequence difference distinguishing serotypes (STs) 1, 2, 4, 5, and 6 from STs 3 and 7 involves a Glu-Lys substitution at position 131 and a Ser-Asn at position 152. These two primary sequence differences conceivably disrupt the Arg-59–Glu-131 salt bridge and the Arg-59 hydrogen bond with Ser-152, consequently reducing the cross-reactivity of Fabs 221-11 and 227-1 with ST3 and ST7. All side chains are drawn as sticks and color coordinated to the main chain color with nitrogen atoms blue and oxygen atoms red. Salt bridges are represented as yellow dashes and hydrogen bonds are red dashes.

throughout the *Borrelia* genus to identify novel OspA types beyond the eight STs and nine ISTs that have been reported [38]. To this end, we constructed an unrooted phylogenetic tree consisting of 135 OspA protein sequences collected from all 23 recognized *Bbsl* genospecies [39]. To facilitate the identification of clades corresponding to known ST/ISTs, we included 86 OspA reference sequences obtained from the PubMLST OspA IST collection assembled by Lee [38].

The resulting phylogenetic tree revealed 33 distinct sequence clusters corresponding to OspA ST 1–8, IST 9–17, and 16 other sequence groups representing previously untyped OspA variants **(Fig 6A)**. The 86 OspA reference sequences grouped into discrete clades consistent with their previously designated ST or IST designations [38]. The one exception was *B. valaisiana*, which clustered into two distinct clades: four sequences clustering as IST15, and three other

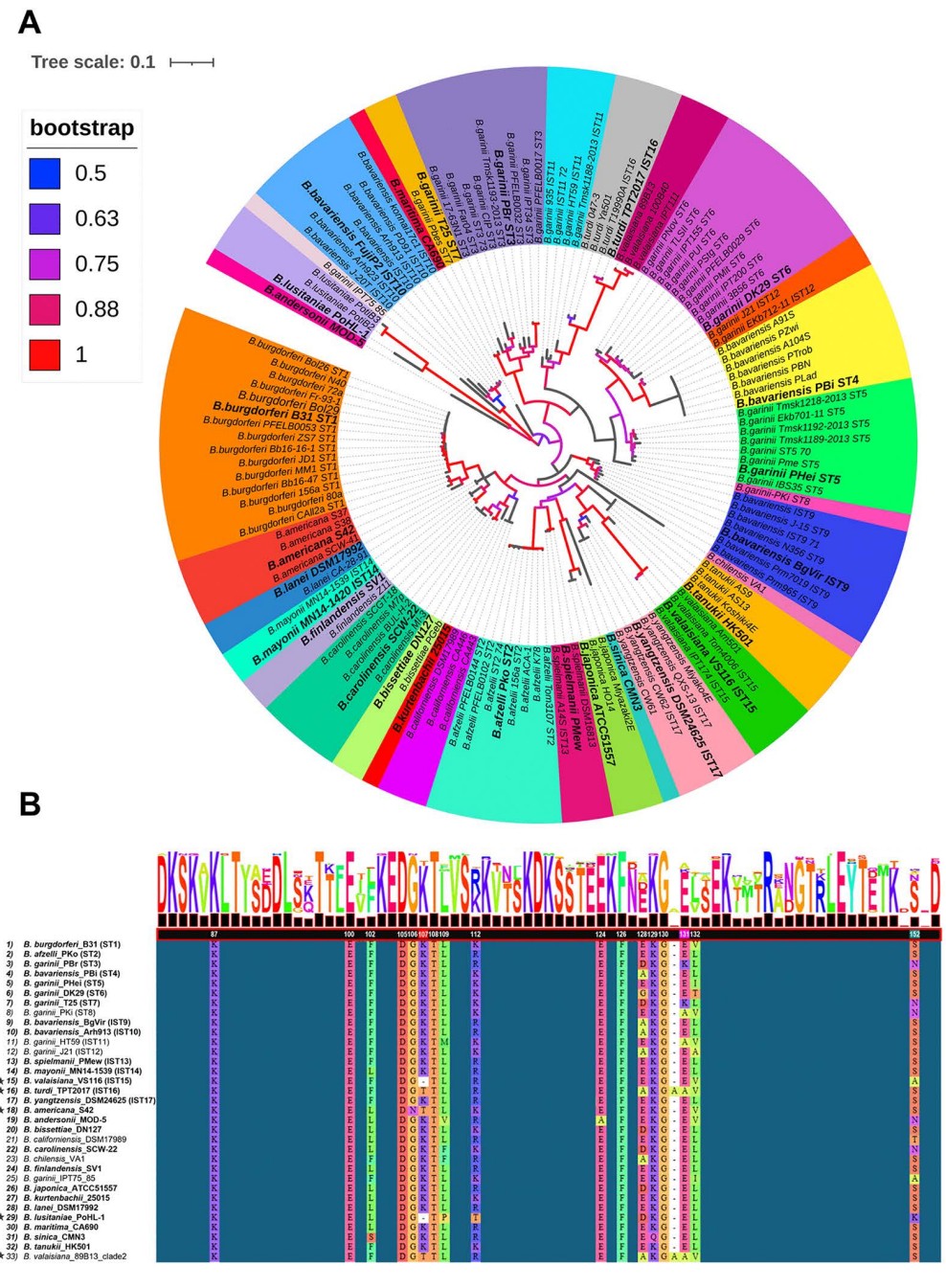

**Fig 6. Phylogenic analysis of OspA within the *Borrelia burgdorferi sensu lato* (Bbsl) genospecies complex. (A)** Unrooted phylogenetic tree comprised of 135 OspA protein sequences from 23 *Bbsl* genospecies. Phylogenetic analysis identified 33 distinct OspA clades corresponding to serotypes (ST) 1-8, *in silico* types (IST) 9-17, and 16 other unclassified OspA IST's. Clades representing each OspA variant/type are distinguished by color. Enlarged and bolded strain names denote OspA alleles selected for HB19-R1 viability reporter strain construction. Bootstrap values representing approximately maximum likelihood analysis are distinguished by color: > 0.50 = green, > 0.75 = blue, 1.00 = red. Nodes with bootstrap values under 0.50 are not depicted in the figure (gray) **(B)** Simplified multiple sequence alignment of the Bin1 epitope region of 33 unique OspA types. Representative sequences from each OspA clade were aligned using OspA from *B. burgdorferi* strain B31 as a reference. Bolded strain names indicate OspA variants expressed by HB19-R1 viability reporter strains. Amino acid residues numbered in white denote conservation within at least two Bin 1 mAb epitopes with solved Fab-OspA crystal structures (857-2, 221-5, 221-11, 221-7, and 227-1). Numbered amino acid residues with colored background form critical interactions with residues within the paratopes of Bin 1 mAbs with solved crystal structures (Lys-107, Glu-31, Ser-152).

sequences sharing close homology with IST16, principally associated with *B. turdi* (**Fig 6A**). We also identified a single *B. garinii* OspA sequence (isolate IPT75) that grouped independently of the eight recognized OspA types found within this genospecies. The 14 other OspA clades identified corresponded to untyped *B. burgdorferi* sl genospecies, including *B. americana, B. andersonii, B. bissettiae, B. californiensis, B. carolinensis, B. chilensis, B. finlandensis, B. japonica, B. kurtenbachii, B. lanei, B. lusitaniae, B. maritima, B. sinica,* and *B. tanukii* (**Fig 6A**).

## Genetically diverse OspA types are susceptible to Bin 1 mAbs

To assess epitope conservation across the *BbsI* genospecies complex, we generated a multiple sequence alignment (MSA) of representative OspA sequences from each of the 33 clades identified in our phylogenetic analysis. Using OspA ST1 as the reference, the alignment revealed broad conservation of the loop regions between β-strands 7–8 and 9–10, including residues associated with 857-2 and other Bin1 mAb epitopes (**Fig 6B** and **S7 Fig**).

To experimentally examine conservation of 857-2's and other Bin 1 epitopes beyond OspA ST 1–7, we generated 19 additional HB19-R1 viability reporter strains expressing OspA variants from two additional *B. bavarienisis* types (IST9, IST10) and 17 other *Bbs*I genospecies. Many of the OspA types selected are occasionally associated with Lyme disease in humans such as *B. bavariensis* IST9 and IST10, *B. spielmanii* (IST13), *B. mayonii* (IST14), *B. americana*, *B. bissettiae*, and *B. lusitaniae* or exhibit limited or uncertain disease potential in humans like *B. andersonii,* and *B. yangtzensis* [39–42]. We also generated HB19-R1 viability reporter strains expressing OspA types from non-pathogenic BbsI genospecies to fully capture BbsI complex-wide sequence diversity and explore the impact of polymorphisms predicted to disrupt Bin 1 mAb binding. These included *B. valaisiana* (IST15), *B. turdi* (IST16), *B. sinica*, *B. finlandensis*, *B. japonica*, *B. lanei*, *B. carolinensis*, *B. kurtenbachii*, *B. maritima*, and *B. tanuki* [39,41,43]. The source strains for each OspA type or IST are indicated in bold within the phylogenetic tree (**Fig 6A**) and corresponding multiple sequence alignment (**Fig 6B** and **S7 Fig**).

The Bin 1 mAbs were then evaluated for complement-dependent borreliacidal activity against the 19 additional OspA reporter strains described above. **Class I** mAbs (221-20, 221-5, 857-2) were the most broadly functional, as exemplified by 221-20, which exhibited potent borreliacidal activity against 23 of the 26 OspA reporter strains within our collection [$EC_{50}$ 1.25-2.50 nM] (**Fig 7**). The three resistant strains ($EC_{50} > 60$ nM) were *B. turdi* (IST16), *B. lusitaniae,* and *B. sinica*. 221-5 and 857-2 susceptibility profiles were similar to 221-20, with the addition of both having lessened activity against *B. valaisiana* (IST15), and 857-2 also being ineffective against *B. americana* (**Fig 7**). The **Class II** mAbs (221-11, 227-2 and 223-5) had borreliacidal activity profiles similar to 857-2, albeit with reduced potencies (**Fig 7**). Interestingly, 223-5 was distinct among this class for its inability to kill *B. carolinensis*. Finally, 227-1 (**Class III**) had borreliacidal activity against just 17 of the 26 OspA reporter strains and reduced efficacy ($Ec_{50}$ 5–10 nM) against OspA variants from *B. mayonii* (IST14), *B. finlandensis, B. japonica,* and *B. lanei* (**Fig 7**). These results demonstrate that one or more borreliacidal epitopes are conserved across virtually all the major variants within the *BbsI* genospecies complex.

## Lys-107 polymorphisms impart resistance to virtually all Bin 1 mAbs

As noted above, HB19-R1 strains expressing OspA from *B. turdi* (IST16) and *B. lusitaniae* PoHL-1 were resistant to all the Bin 1 mAbs tested, while *B. americana* and *B. valaisiana* (IST15) were resistant to a subset of **Class I** mAbs and all the **Class II** and **Class III** mAbs (**Fig 7**). Examination of the OspA MSA (**Fig 6B**) suggested that polymorphisms at $OspA_{ST1}$ Lys-107 may account for all or part of the observed resistance. In the case of *B. turdi* (IST16), for example, there is a Thr residue rather than a Lys at position 107, while OspA from *B. lusitaniae* has a deletion at the equivalent position (**Fig 6B**). Polymorphisms at Lys-107 would preclude 857-2 and other mAbs from forming critical salt bridges with OspA (**Fig 4**).

To investigate the importance of Lys-107 in the context of OspA recognition by the Bin 1 mAbs, we generated HB19-R1 viability reporter strains expressing $OspA_{ST1}$ variants harboring a K107T substitution or Lys-107 deletion (ΔK107). Compliment-dependent killing assays revealed that the $OspA_{ST1}$ K107T and ΔK107 reporter strains were completely resistant to **Class II** (227-2, 223-5, 221-11) and **Class III** mAbs (227-1), even at the highest antibody concentrations tested

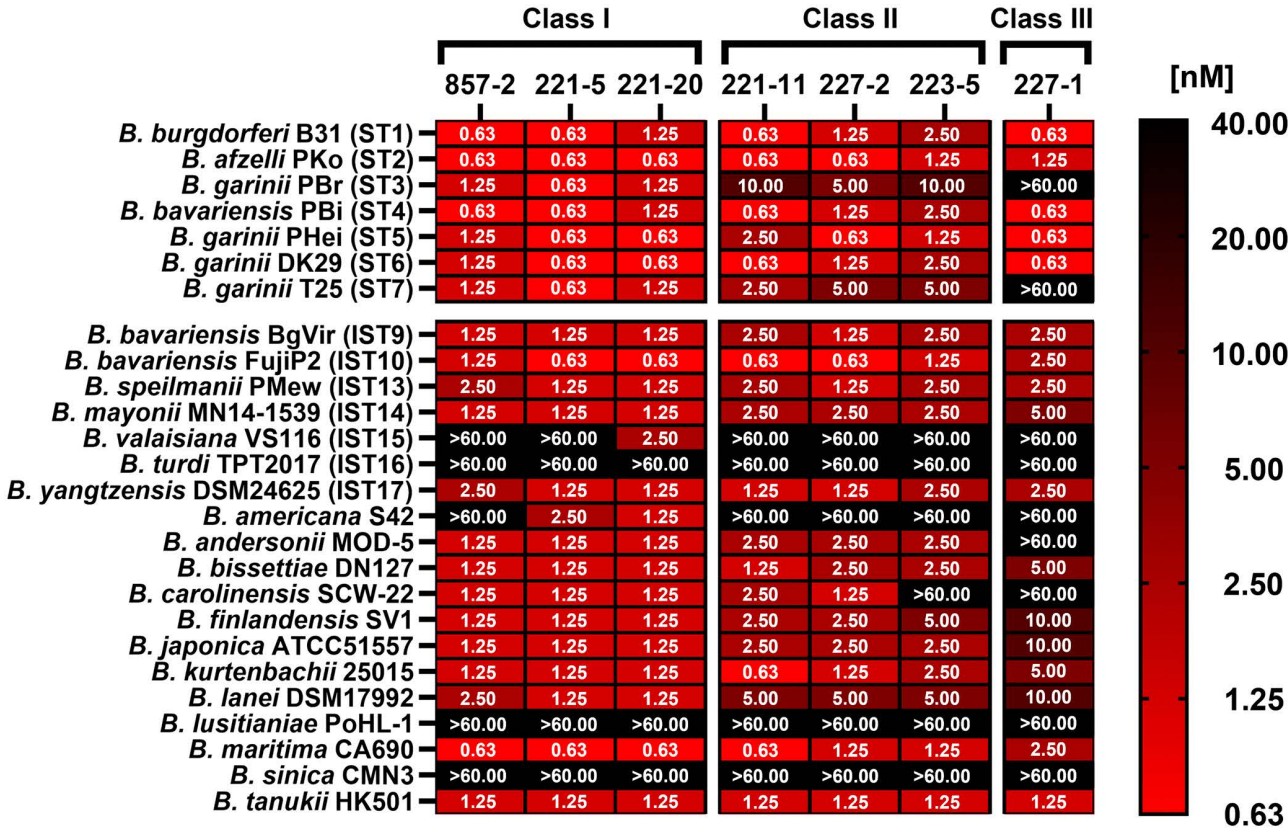

**Fig 7. Genetically diverse OspA types are susceptible to complement-mediated killing by anti-OspA_{ST1} Bin 1 mAbs.** Complement-dependent bactericidal assays were performed as described in Materials and Methods using anti-OspA Bin 1 mAbs and a panel of *B. burgdorferi* HB19-R1 *mScarlet-I* viability reporter strains expressing 19 OspA *in silico* types (ISTs). The heat map summarizes mean EC_{50} values for complement-mediated killing by Bin 1 mAbs from Classes 1-3. EC_{50} values represent the lowest mean antibody concentration (nM) producing >50% reduction in *mScarlet-I* fluorescence relative to untreated controls and were derived from ≥3 independent experiments. Susceptibility profiles of reporter strains expressing OspA ST1-7 are included for comparison. Strains that exhibited resistance to mAbs at 10 nM were retested at 66.6 nM. Corresponding antibody titration curves and statistical comparisons between the HB19-R1 OspA ST1 reporter strain and strains expressing OspA ST2-7 are shown in **S8 Fig**.

(66.6 nM) (**Fig 8**). Similarly, both modifications were detrimental to **Class I** mAbs 857-2 and 221-5's borreliacidal activity, as evidenced by 16- fold increases in EC_{50} values (**Fig 8**). For 857-2, the OspA_{ST1} ΔK107 deletion was slightly more deleterious than the K107T substitution (EC_{50} 20 nM vs 5 nM), while 221-5's resistance was comparable between the two (10 nM). Conversely, the OspA_{ST1} K107T and ΔK107 reporter strains remained sensitive to killing by 221-20, demonstrating the unique nature of 221-20's paratope-epitope interaction and differentiating 221-20 from the other two Bin 1 mAbs.

In light of these results, we examined whether introduction of Lys-107 in an otherwise resistant OspA variant from *B. valaisiana* (IST15) was sufficient to render cells susceptible to killing by the Bin 1 mAbs. In accordance with that hypothesis, the OspA_{IST15}—G106insK reporter strain was susceptible to all Bin 1 mAbs with EC_{50} values comparable to OspA_{ST1} (EC_{50} range 0.63- 2.50 nM), thereby demonstrating that antibody recognition is transformed by the addition of a single Lys residue at position 107. We also generated a reporter strain with a *B. turdi* OspA (IST16) allele encoding a T107K substitution to investigate the role of Lys-107 in the context of polymorphisms elsewhere in 857-2 and other Bin 1 mAb epitopes (e.g., residues 128–132). In fact, the T107K substitution was sufficient to render *B. turdi* OspA at least partially susceptible to the **Class I** mAbs (857-2, 221-5, 221-20) and one **Class II** mAb (223-5) with EC_{50} values ranging from 10-40 nM (**Fig 8**). Collectively, these experiments demonstrate that Lys-107 is a key susceptibility determinant for virtually all Bin 1 mAbs.

**A**

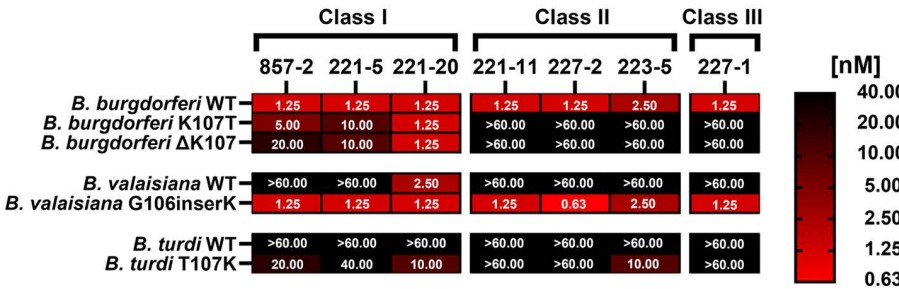

**B**

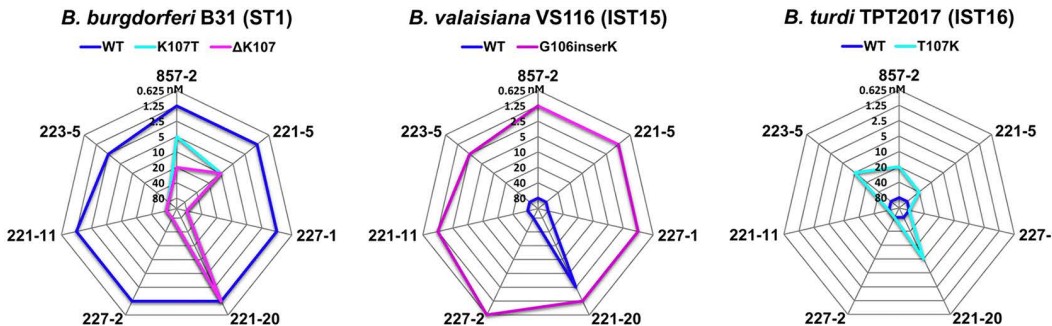

**C**

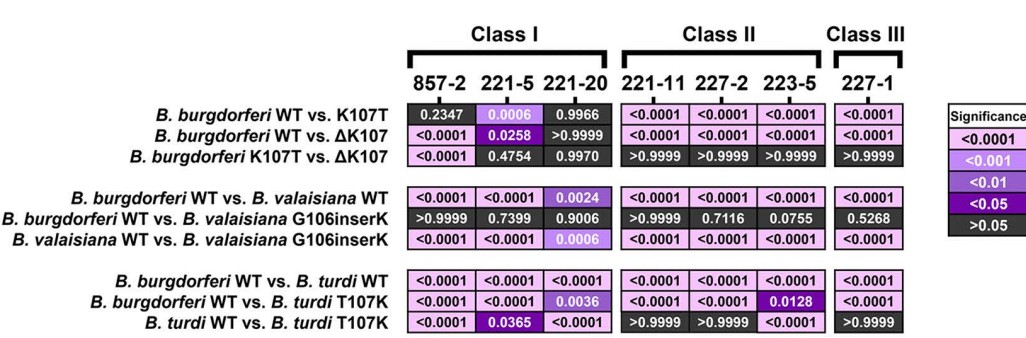

**Fig 8. Lys-107 polymorphisms are a determinant of susceptibility to nearly all Bin 1 mAbs** Complement-dependent killing assays were performed as described in the Materials and Methods with anti-OspA Bin1 mAbs and *B. burgdorferi* HB19-R1 viability reporter strains expressing "WT" OspA variants from *B. burgdorferi* B31 (ST1), *B. valaisiana* VS116 (IST15), and *B. turdi* TPT2017 (IST16), or mutated derivatives containing deletions, insertions, or substitutions at residue 107. Heat maps **(A)** and radar plots **(B)** depicting the Ec$_{50}$ profiles of WT and mutated OspA variants. Ec$_{50}$ values denote the lowest antibody concentration (nM) capable of reducing mScarlet-I fluorescence by >50% relative to untreated controls and represent the mean of > 3 independent experiments. Reporter strains exhibiting resistance to killing by mAbs at 20 nM were re-examined at 40 nM and 66.6 nM doses. **(C)** Heat maps graphically depicting significant differences in mAb susceptibility between HB19-R1 reporter strains expressing WT and mutated OspA variants as determined via two-way ANOVA followed by Tukey's multiple comparisons test. Adjusted P-values reflecting significant differences between strains are shown in shades of purple or pink. Antibody titration curves summarizing borreliacidal activity against each strain are provided in **S9 Fig**.

## Discussion

The development of cross-protective vaccines for Lyme disease has been challenging due to the antigenic variability of OspA among the major pathogenic *B. burgdorferi* genospecies in North America, Europe, and Asia. Efforts to identify conserved epitopes on OspA have been limited, despite evidence in the literature for their existence [17,44]. Wilske and colleagues, for example, described a mouse mAb (L31 1F11) capable of reacting with all seven OspA STs and more than 100 primary isolates [15]. L31 1F11's epitope was localized through a series of OspA truncations and Western blotting to residues 102–111, which corresponds to OspA's central β-sheet [15]. Twenty years later, Wang and colleagues "rediscovered" this region of OspA in their effort to identify broadly reactive human mAbs with the potential to serve as preexposure prophylactics for Lyme disease [11]. Indeed, in our current study we identified overlapping protective B cell epitope(s) that are shared across all the major OspA serotypes associated with Lyme disease-causing spirochetes. The core of those epitopes involves residues within $OspA_{ST1}$ β-strands 6–10 and loops 7–8 and 9–10 (**S10 Fig**). Within this region, there are invariant (e.g., Glu-100, Asp-105) and highly conserved residues responsible for cross-serotype recognition. Varying paratope interactions within and outside this core epitope account for the different degrees of mAb reactivity across the seven major OspA serotypes. For example, single amino acid polymorphisms relative to $OspA_{ST1}$ accounts for the reduced binding of 221-11 (**Class II**) and 227-1 (**Class III**) to $OspA_{ST3}$ and $OspA_{ST7}$. Nonetheless, these results suggest that it may be possible to design a "universal" monovalent Lyme disease vaccine antigen focused on OspA's central β-sheet as an alternative strategy to multivalent chimeric vaccines derived from the variable C-terminal regions of the major OspA serotypes [23–26].

Also relevant to vaccine immunology, we observed a predilection of particular $V_H$ and VL combinations for recognition of OspA's central β-sheet. Namely, the seven Bin 1 mAbs characterized in this study as well as 221-7 are predicted to utilize $V_H$ germline HV5–51 with critical epitope interactions mediated largely (if not exclusively) by germline encoded residues. Similarly, critical light chain contacts are associated with KV1–13 or KV3–11 germline residues, suggesting that particular BCR pairings (i.e., HV5–51/ KV1–13) may be sufficient to engage with conserved residues within OspA's central β-sheet, including Lys-107. However, as these Bin 1 mAbs were generated from hybridomas derived from hyperimmunized transgenic mice containing human immunoglobulin genes [11], we can only speculate that such interactions are clinically relevant. BCR repertoire analysis and the role of unmutated, class switch B cells and particular germline in response to Lyme disease has garnered attention recently [45–47]. Jiang and colleagues examined BCR usage and frequency of somatic hypermutation in memory B cell populations from erythema migrans skin lesions and noted a subpopulation of unmutated memory B cells [46]. Blum and colleagues examined BCR repertoires in circulating plasmablasts and, perhaps coincidently identified HV5–51 as being one of just two $V_H$ germlines overrepresented in Lyme disease patients, as compared to healthy controls [45].

The generation of isogenic *B. burgdorferi* reporter strains expressing the seven major OspA serotypes (ST1–7) and numerous ISTs proved instrumental in being able to define cross-borreliacidal epitopes recognized by the seven mAbs in this study. We speculate that the same reporters would be equally valuable in assessing antibody responses elicited by OspA-based vaccines. At present, assessing complement-dependent bactericidal activity across different Bbsl genospecies requires the development of genospecies-specific assays in which culture conditions, complement sources, and assay duration must be optimized for each individual strain [48,49]. For example, because of *B. garinii*'s (ST3) inherent sensitivity to guinea pig, mouse and human complement, Comstedt and colleagues had resorted to freshly isolated chicken blood for bactericidal assays [25]. Moreover, the overwhelming reliance on ATP-based viability assays is problematic due to the nature of *B. burgdorferi* metabolism and high signal to noise ratios. The adoption of a standardized fluorescence-based, serum borreliacidal assay (SBA) based on OspA reporter strains like those described herein would not only streamline the assay itself but would enable different manufacturers and regulatory agencies to compare vaccine efficacies using a single platform.

## Materials and methods

### Cloning and expression of OspA and anti-OspA mAbs

Recombinant human monoclonal IgG1 antibodies were purified via Protein A affinity column from Expi293 cells that had been transiently transfected with $V_H$ and $V_L$ plasmids, as described [11]. Recombinant $OspA_{ST1}$ (residues 18–273) from *B. burgdorferi* B31 (NCBI:txid224326) was expressed in *Escherichia coli* BL21-DE3 and purified as described [12]. Purified OspA was maintained in 20 mM HEPES, 150 mM NaCl, 20 mM imidazole, pH 7.50. Recombinant non-lipidated OspA serotypes 1–7 purified from *E.coli* were kindly provided by Dr. Meredith Finn (Moderna) and described elsewhere [34].

### ELISA

Immulon 4HBX plates (ThermoFisher, Waltham, MA) were coated overnight at 4°C with 1 µg/mL of the seven OspA serotypes (non-lipidated) and the next morning blocked for 2 h with 2% goat serum in PBS supplemented with 0.1% Tween-20 (PBST). Serial two-fold dilutions of each antibody starting from 1 µg/mL were made in separate dilution plates, and transferred to the serotype coated plates for 1 h. Plates were then washed three times with PBST, and a goat anti-human HRP conjugated secondary antibody (SouthernBiotech) was added for 1 h. Plates were then washed three times with PBST, developed with SureBlue TMB substrate (Seracare, Milford, MA), and quenched with 1M phosphoric acid. Absorption at 450 nm was read by a SpectraMax iD5 spectrophotometer using Softmax Pro v7.1 (Molecular Devices, San Jose, CA).

### Affinity measurements with BioLayer interferometry

Certain monoclonal antibody binding kinetics first reported by Haque and colleagues [32] have been revised to reflect refined and optimized BLI protocols using an Octet RED96e (Sartorius AG, Gottingen, Germany). AHC sensors (#18–5060, Sartorius) were used to capture mAbs at 0.4 µg/mL for 5 min, to a loading level of ~0.5 nm, then dipped into wells containing a three-fold dilution series ofnon-lipidated OspA (ranging from 0.6 µg/mL, 21.58 nM to 0.0025 µg/mL, 0.089 nM) for 10 min. Finally, sensors were dipped into buffer alone for 30 minutes to measure dissociation. Buffer used in all wells was 0.1% BSA in PBS, with 0.01% Tween20. Sensors were regenerated before the first mAb and between each mAb with 0.2 M glycine, pH 2.2. Traces were background corrected with a reference sensor, in which mAb was loaded but not exposed to OspA, and a reference sample, in which a bare sensor was exposed to the highest concentration of OspA. Traces for each mAb were fit to a global 1:1 model. $k_{obs}$ vs concentration was linear for each system.

### Construction of *B. burgdorferi* strains expressing OspA serotypes

The IPTG-inducible *mscarlet-I* viability reporter plasmid, pGW189 [34] was modified to facilitate the expression of *ospA* variants of interest under the regulatory control of the $P_{ospAB}$ promoter from *B. burgdorferi* strain B31. To accomplish this, transcription terminators were first introduced into pGW189 to prevent read through transcription from the *lacI* and *mscarlet-I* ORFs. The lambda_T0_rrnBT1 and rrnbT2 transcription terminators were amplified from plasmids pTN7 [50] and puc18_mtn7-lacZ_gentR [51] using NEB's Q5 DNA polymerase and tailed primers listed in **S5 Table**. Amplified DNA fragments were then assembled into pGW189 immediately downstream of the *lacI* and *mscarlet*-I ORFs using the SacI/HindIII restriction sites and NEB's HiFi Assembly kit, creating pGW206.

The *ospA* promoter and ORF from *B. burgdorferi* B31 were next amplified in two overlapping DNA fragments with tailed primers designed to introduce a silent SphI restriction site ~50 bases into the *ospA* ORF and permit assembly into pGW206 via the PvuI and PvuII restriction sites. The resulting viability reporter/OspA ST1 expression plasmid, pGW217, subsequently served as the basis for creating additional OspA expression plasmids. To accomplish this, pGW217 was first digested with SphI and PvuII to remove the $ospA_{ST1}$ ORF immediately following the conserved lipoprotein insertion sequence (amino acid residue 17). Gene fragments from additional *ospA* serotypes/*Bbsl* genospecies were then synthesized (IDT) and assembled directly into the vector using HiFi DNA Assembly kit (NEB).

Silent mutations were introduced into *ospA* gene sequences when necessary to reduce sequence complexity for *in vitro* synthesis. Mutations were introduced at sites that resulted in a shift to the most common or second most common codon utilized by *B. burgdorferi* B31. The codon adaptive index for each *ospA* cDNA was assessed prior to synthesis using JCAT [52] to ensure that the introduced silent mutations did not reduce translation efficiency in *B. burgdorferi.*

Gene fragments were also synthesized to create *B. burgdorferi* B31, *B. valaisiana,* and *B. turdi* reporter plasmids with insertions, deletions, or substitutions at position 107 within the *ospA* ORF. To permit direct comparison with *B31* point mutants, a second "wild type" OspA ST1 reporter plasmid was also created containing the same silent mutations that were introduced to increase GC content/reduce sequence complexity for *in vitro* synthesis. The sequences of primers and synthesized *ospA* gene fragments used in this study are provided in **S5** and **S6 Tables**.

OspA reporter plasmids were sequenced and transformed into *B. burgdorferi* HB19-R1 [37], a high-passage derivative of the human blood isolate HB19 that lacks linear plasmid 54 (*ospAB*-), using previously described methods [53]. Transformants were selected in liquid BSKII supplemented with gentamicin (100 µg/ml) and individual clones were isolated through serial dilution [53]

To evaluate native plasmid content in transformants, the plasmid profiles of *B. burgdorferi* HB19 and HB19-R1 were characterized using NEB's Taq Quickload 2×PCR MM with established *B. burgdorferi* plasmid profiling primer sets [54]. Additional primer sets targeting cp26, lp21, lp28–1, lp5, lp28–5, and lp28–6 were also included in the analysis [54,55].Since plasmid content was uniform among clones transformed with mScarlet-I reporter plasmids carrying *ospA* ST1–7 (pGW217–pGW223, pGW250), plasmid profiling was not performed on HB19-R1 reporter strains encoding OspA ISTs due to the labor-intensive nature of the screening process. All *Bbsl* strains used or constructed in this study are listed in **S1** and **S2 Tables**.

## Flow cytometry-based antibody binding to strains

To analyze the ability of Bin 1 mAbs to bind to native OspA ST1–7 on the bacterial surface of primary isolates, *Borrelia* strains *B. burgdorferi* B31 (ST1), *B. afzelii* Pko (ST2), *B. garinii* PBr (ST3), *B. bavariensis* PBi (ST4), *B. garinii* PHei (ST5), *B. garinii* TN (ST6), and *B. garinii* T25 (ST7) (kindly provide by Dr. John Leong, Tufts University) were cultured at 33°C with 5% $CO_2$ until mid-log phase. Bacterial cells were stored at -80ºC in fresh media containing 20% glycerol, and preparation, incubations, and flow cytometry analysis was performed as described [47]. To analyze the ability of Bin 1 mAbs to bind to OspA ST1–7 on the bacterial surface of reporter strains HB19-R1, (GGW1072–1079; **S2 Table**), bacterial cells were cultured at 33°C with 5% $CO_2$ in BSKII media, minus gelatin and containing gentamycin 50 µg/ml, until mid-log phase. Bacterial cell preparation, incubations, and flow cytometry analysis was performed as described [56]. The *Salmonella*-specific mAb, Sal4 IgG, was used as an isotype control [57].

## Complement-dependent borreliacidal assays

Complement-dependent bactericidal assays were performed as previously described [34] with several modifications. Because guinea pig and human-derived complement perform equivalently in this assay with respect to sensitivity and reproducibility (**S11 Fig**), all experiments were conducted using guinea pig complement due to financial constraints. Frozen aliquots of OspA-expressing HB19-R1 reporter strains were thawed at room temperature and 500 µl from each stock (equivalent to $5 \times 10^7$ spirochetes) were transferred to 50 ml conical tubes containing 45 ml of in gelatin-free BSK II supplemented with 50 µg/ml of gentamicin. Tubes were then sealed, and the cultures were incubated at 33ºC under static growth conditions. Three days later, spirochetes were collected via centrifugation (4,000 x *g*), the supernatant was removed, and cell pellets were resuspended in phenol red-free BSK II with gentamicin (50 µg/ml) at a density of $3 \times 10^7$ spirochetes/ml.

Spirochetes were next seeded into white bottom 96 well microtiter assay plates (Costar) and mixed 1:1 with serial dilutions of anti-OspA$_{ST1}$ mAbs prepared in phenol red-free BSKII containing 5% guinea pig complement (Sigma-Aldrich) and gentamicin (50 µg/ml), resulting in $3 \times 10^6$ spirochetes per well with a final complement concentration of 2.5% and antibody concentrations between 10.00 nM - 0.08 nM, or 20 nM - 0.16 nM (depending on strain/experiment. Eight untreated

(antibody-free) control reactions per strain were also included on each assay plate to determine baseline and peak fluorescence for data normalization.

Following antibody addition, assay plates were incubated overnight at 37ºC with 5% $CO_2$ without agitation. The next day (18–20 h later), 1 mM IPTG was added to appropriate wells to induce expression of *mScarlet* in surviving spirochetes. Assay plates were then returned to the incubator for another 24 h period. The following day, median fluorescence intensity (MFI) was measured three times per plate at 569 nm (excitation) and 611 nm (emission) using a Spectromax ID3 plate reader (Molecular Biosystems) and Softmax pro version 7.1 software.

Mean MFI data was independently normalized for each assay plate using the untreated/non-induced and untreated/IPTG-induced controls to establish baseline (0) and peak (1) MFI. Graphpad Prism version 9.5.1 was then used to generate killing curves and heatmaps, while radar plots were constructed in Microsoft Excel. $Ec_{50}$ values were assigned to the lowest antibody dilution resulting in >50% reduction in MFI relative to untreated controls. $Ec_{50}$ values reported in heatmaps and radar plots reflect the mean calculated using normalized MFI data from three to five independent experiments. Strains that exhibited resistance to anti-OspA Bin 1 mAbs at the highest concentration tested (10 or 20 nM) were subjected to additional bactericidal assays with 10 µg/ml of the antibodies in question. Resistance at this dosage is shown as ">60 nm" in heat maps and radar plots.

## Cloning, expression and purification of OspA and anti-OspA Fabs

The PCR amplicon encoding *B. burgdorferi* OspA residues 18–273 was subcloned in frame into the pSUMO expression vector with an N-terminal deca-histidine and SUMO tag. All cloning was performed using a standard ligase independent cloning protocol. OspA was expressed in *E. coli* strain BL21 (DE3). The transformed bacteria were grown at 37°C in TB medium and induced at 20°C with 0.1 mM (IPTG) at an $OD_{600}$ of 0.6 for ~16 hours at 20°C. After induction, cells were harvested and resuspended in 20 mM Hepes pH 7.5 and 150 mM NaCl. The cell suspension was sonicated and centrifuged at 30,000 g for 30 minutes. After centrifugation, the protein-containing supernatant was purified by nickel-affinity and size-exclusion chromatography on an AKTAxpress system (GE Healthcare), which consisted of a 1mL nickel affinity column followed by a Superdex 200 16/60 gel filtration column. The elution buffer consisted of 0.5M imidazole in binding buffer, and the gel filtration buffer consisted of 20mM Hepes pH 7.5, 150mM NaCl, and 20mM imidazole. Fractions containing OspA (18–273) was pooled and subject to TEV protease cleavage (1:10 weight ratio) for 3 hours at room temperature in order to remove their respective fusion protein tags. The cleaved protein was passed over a 1mL Ni-NTA agarose (Qiagen) gravity column to remove the added TEV protease, cleaved residues, and uncleaved fusion protein. Each IgG was subjected to papain digestion followed by affinity depletion of the Fc fragment by Protein A FPLC chromatography to generate each Fab. After purification, each Fab and OspA were mixed in a 1:1 stoichiometry to form a stable complex concentrated to a final concentration of 10 mg/ml for all crystallization trials.

## Crystallization and data collection

Fab-OspA crystals were grown by sitting drop vapor diffusion at 4°C using a protein to reservoir volume ratio of 1:1 with total drop volumes of 0.2 µl. Crystals of each OspA-Fab complex were produced using the crystallization solutions shown in **S3 Table**. All crystals were flash frozen in liquid nitrogen after a short soak in the appropriate crystallization buffers supplemented with 25% ethylene glycol. Data were collected at the 24-ID-E beamline at the Advanced Photon Source, Argonne National Labs. Data resolution limit was determined based on the I/s criterion which was 2.0 in the highest resolution shell. All data was indexed, merged, and scaled using HKL2000 [58] then converted to structure factor amplitudes using CCP4 [59].

## Structure determination and refinement

The structure of each Fab-OspA complex was solved by molecular replacement using Phaser [58]. Molecular replacement calculations were performed using the Fab coordinates from PDB code 2XTJ as the search model for Fab 221-5, the Fab

heavy chain coordinates from PDB code 4RIR along with the Fab light chain coordinates from PDB code 4M6O were independently used as search models for Fabs 221-11 and 857-2. The Fab coordinates from PDB code 4M6O were also used as the search model for Fab 227-1. OspA coordinates (PDB code 1FJ1) were used as the search model for OspA in all four Fab-OspA complex structures solved. The resulting phase information from molecular replacement was used for manual model building of the Fab-OspA model using the graphics program COOT [60] and structural refinement employing the PHENIX package [61]. Data collection and refinement statistics are listed in **S4 Table**. Molecular graphics were prepared using PyMOL (Schrodinger) (DeLano Scientific LLC, Palo Alto, CA). The structures generated in this study were deposited in the Protein Data Bank (PDB; http://www.rcsb.org/pdb/) under accession numbers shown in **Table 2**.

### Phylogenetic tree construction

A collection of 135 OspA protein sequences containing representatives from 23 recognized BbsI genospecies was compiled from borreliabase.org, NCBI's OspA IST typing database (PubMLST), and Bacterial and Viral Bioinformatics Resource Center (BV-BRC.org). MUSCLE (accessed via EMBL) was then used to generate a multiple sequence alignment of the collection. The resulting MSA file was then used to construct an unrooted approximately maximum likelihood phylogenetic tree using Fasttree and LG modelling (accessed via BV-BRC.ORG). The resulting phylogenetic tree was then formatted and visualized using ITOL version 7 [62]. Files associated with phylogenetic tree construction are provided in **S1 File**.

### OspA multiple sequence alignment (MSA)

A MSA was created containing one representative sequence from each of the 33 OspA clades/types identified through phylogenetic analysis. OspA variants selected for the alignment included OspA sequences utilized in viability reporter strain construction and OspA sequences from type or reference strains (when available). Sequence alignment was performed using the MAFFT algorithm with OspA from *B. burgdorferi* B31 as a reference. Following alignment, the Bin 1 epitope region (β strands 6–10, residues 82–146) was visualized using the MSA viewer interface on BV-BRC.org. A graphic of the alignment was then created with the Taylor color scheme and the sequence logo displayed to emphasize conservation between OspA types/clades.

### Microsphere immunoassay (MIA)

ST 1–7 and OspB were coupled to Magplex-C microspheres (5mg antigen/ $1\times10^6$ microspheres) using a xMap Antibody Coupling Kit as recommended by the manufacturer (Luminex Corporation, Austin, TX). Successful coupling of ST 1–7 and OspB was confirmed using LA-2 and H6831, respectively. Beads were protected from light and stored 2–8°C in xMAP AbC Wash Buffer ($5\times10^6$ microspheres/mL) until use.

OspA Bin 1 mAbs were serially diluted (1:2) in assay buffer (1 x PBS, 2% BSA, pH 7.4) in round bottom, non-treated plates (Costar, Kennebunk, Maine) with starting concentrations of 2.5 mg/mL (223-5), 0.625 mg/mL (857-2, 221-5, 221-20, 221-11, 227-1), or 0.3125 mg/mL (227-2). Coupled microsphere stocks were diluted (1:50) in assay buffer and added (50 μL/well) to black, clear-bottomed, non-binding, chimney 96-well plates (Greiner Bio-One, Monroe, North Carolina). The mAb dilutions (50 μL) were combined with the microspheres and incubated at room temperature for 1 hr in a tabletop shaker (600 rpm). Plates were placed on a magnetic separator and washed three times using wash buffer (1 x PBS, 2% BSA, 0.02% TWEEN-20, 0.05% Sodium azide, pH 7.4). PE labeled goat anti-Human IgG Fc, eBioscience (Invitrogen, Carlsbad, California) secondary antibody diluted 1:500 in assay buffer was added (100 μL/well) and incubated at room temperature for 30 min in a tabletop shaker (600 rpm). Plates were washed as previously stated. The microspheres were resuspended in 100 μL of wash buffer and placed back on the tabletop shaker (600 rpm) for 5 minutes prior to analysis using a FlexMap 3D (Luminex Corporation).

## Supporting information

**S1 Table.** *Borrelia burgdorferi sensu lato* **strains used in this study.**
(DOCX)

**S2 Table. Recombinant** *B. burgdorferi* **strains used in this study.**
(DOCX)

**S3 Table. OspA-Fab crystallization solutions.**
(DOCX)

**S4 Table. Summary of OspA-Fab structures in this study.**
(DOCX)

**S5 Table. PCR primers used in this study.**
(DOCX)

**S6 Table. Sequences of** *ospA* **gene fragments synthesized for reporter construction.**
(DOCX)

**S1 Fig. 857-2 binds to OspA serotypes 1–7 expressed on the surface of** *Borrelia* **primary isolates.** Representative flow cytometric analysis of live *Borrelia* primary isolates expressing OspA serotypes (ST) 1–7 with 10 µg/ml of Bin 1 IgG mAb 857-2, followed by addition of Alexa 647-labeled goat anti-human IgG secondary antibody. An isotype control, O5 antigen-specific IgG variant Sal4, was run with each ST, however ST 1 is shown as the representative. OspA ST 1-specific IgG mAb LA-2 was run with ST 1–7 for comparison. The horizontal bracket represents the region on subsequent plots that are positive for fluorescence labeling (647+). The percentage of events positive for Alexa 647 fluorescence labeling and the geometric mean fluorescence intensity (gMFI) are shown. Compared to isotype controls, gMFI values of LA-2 bound to ST1, and 857-2 bound to ST1–7 are significantly different ($p < 0.0001$). There is no significant difference in gMFI with LA-2 and ST2–7 strains ($p > 0.9999$). N = 3. Statistical comparisons were performed using two-way ANOVA with Dunnet's multiple-comparison test. Image was generated in FlowJo. ST 1-*B. burgdorferi* B31; ST 2- *B. afzelii* Pko; ST 3- *B. garinii* PBr; ST 4- *B. bavariensis* PBi; ST 5-*B. garinii* PHei; ST 6- *B. garinii* TN; ST 7- *B. garinii* T25.
(PDF)

**S2 Fig. Binding of Bin 1 mAbs to recombinant OspA ST1–7.** Bin 1 IgG mAbs were subjected to MIA using non-lipidated recombinant ST 1–7 coupled microspheres as detailed in the materials and methods. Recombinant OspB was included as a negative control. Median fluorescent intensity (MFI) is displayed on the y-axis and mAb concentration is shown on the x-axis. The plotted lines show the mean MFI and standard deviation for each serotype across 3 technical replicates with each point representing a single experiment. The concentrations shown are two-fold serial dilutions with starting concentrations of 2.5 µg/mL (223-5), 0.625 µg/mL (857-2, 221-5, 221-20, 221-11, 221-7, 227-1), or 0.3125 µg/mL (227-2).
(PDF)

**S3 Fig. Anti-OspA$_{ST1}$ Bin1 mAbs promote complement-dependent killing of recombinant** *B. burgdorferi* **Strains expressing OspA ST1–7.** (A) Antibody titration curves depicting bactericidal activity against *B. burgdorferi* HB19-R1 reporter strains expressing OspA$_{ST1–7}$. Complement-dependent bactericidal assays were performed with anti-OspA$_{ST1}$ Bin1 mAbs and *B. burgdorferi* HB19-R1 strains harboring an IPTG-inducible mscarlet-I viability reporter plasmid expressing *ospA* serotypes 1–7 as described in the material and methods. Experimental controls included an HB19-R1 strain carrying the IPTG-inducible viability reporter plasmid without an *ospA* variant and a mAb with bactericidal activity restricted to OspA$_{ST1}$ (LA-2). The data shown encompasses 3–5 independent experiments per strain with data normalized as described

within the materials and methods section. (B) Heat map summarizing statistical comparison of differences in susceptibility to anti-OspA ST1 mAbs between the HB19-R1 OspA$_{ST1}$ reporter strain and reporter strains expressing OspA$_{ST2-7}$. Statistical analyses were performed via one-way ANOVA followed by Dunnett's multiple-comparison test. Significant differences in susceptibility are denoted by purple or pink shading.
(PDF)

**S4 Fig. Structural similarity of OspA.** Superpositioned Cα-traces of the unliganded form of OspA (PDB ID: 2G8C) colored green with OspA form all four Fab-OspA structures (shaded from light to dark gray) depicting the structural similarity of OspA across these different structures.
(PDF)

**S5 Fig. OspA sequence alignments.** The high sequence similarity between OspA residues in *B. burgdorferi* (ST1) that interact with Fabs 857-2, 221-5, 221-11, 227-1 and the OspA sequence conservation from serotypes 2–7, supports our observed cross-reactivity of mAbs 857-2, 221-5, 221-11, 227-1 across nearly all seven serotypes. Sequence alignment of the interacting regions of OspA from *B. burgdorferi* (ST1) with (A) 857-2 (B) 221-5 (C) 221-11 (D) 227-1 aligned with OspA sequences ST2-ST7. The yellow boxes in all four panels highlight residue positions 131 and 152, which mark primary sequence differences among OspA serotypes that influence antibody cross-reactivity. In ST 3 and 7, residue 131 is a lysine (Lys-131) rather than a glutamate (Glu-131), and residue 152 is an asparagine rather than a serine. These substitutions appear to underlie the reduced cross-reactivity of Fabs 221-11 and 227-1 with ST3 and ST7. Magenta asterisks depict VL interacting residues, cyan asterisks depict VH interacting residues, and blue asterisks denote interaction with VL and VH residues. Secondary structural elements (β-strands 4–14 in panels A,B,C and β-strands 4–19 in panels D) are drawn as black arrows above the sequence and labelled accordingly. Red colored sequence denotes sequence identity and white colored sequence shows reduced sequence similarity. Figure made with ClustalW and ESPript 3.0.
(PDF)

**S6 Fig. Inferred germline usage.** (A) Heavy chains of the 8 Bin 1 mAbs aligned to the HV5–51*01 Germline Gene. (B) Light chains of 221-5 and 221-20 aligned to KV3–11*01 Germline Gene. (C) Light chains of the other six mAbs aligned to KV1–13*02 Germline Gene.
(PDF)

**S7 Fig. Multiple sequence alignment depicting total sequence diversity within the Bin1 epitope region of 33 OspA types.** Representative sequences from each OspA clade were aligned via MAFFT using OspA from *B. burgdorferi* strain B31 as a reference. Bolded strain names indicate OspA variants expressed by HB19-R1 viability reporter strains and subjected to complement-dependent bactericidal assays with anti-OspAST1 Bin 1 mAbs. Amino acid residues numbered in white font denote conservation within the epitopes of two or more Bin1 mAbs (857-2, 221-5, 221-11, 221-7, and 227-1). Numbered amino acid residues with colored background form critical interactions with residues within the paratopes of Bin1 mAbs with solved crystal structures.
(PDF)

**S8 Fig. Genetically diverse OspA types are susceptible to complement-mediated killing by anti-OspA$_{ST1}$ Bin1 mAbs.** (A) Antibody titration curves depicting complement-dependent killing of *B. burgdorferi* HB19-R1 reporter strains expressing OspA *in silico* types (ISTs). Complement-dependent killing assays were performed as described in the materials in methods section with anti-OspA Bin1 mAbs and HB19-R1 viability reporter strains expressing diverse OspA ISTs. The data shown encompasses at least three independent experiments per strain with data normalized as described. (B) Heat map summarizing statistical comparison of differences in Bin1 mAb susceptibility between the HB19-R1 OspA$_{ST1}$-expressing reporter strain and reporter strains expressing all other OspA variants as determined via one-way ANOVA

followed by Dunnett's multiple-comparison test. Significant differences in susceptibility relative to OspA$_{ST1}$ are denoted by purple or pink shading.
(PDF)

**S9 Fig. Lys-107 polymorphisms are a determinant of Bin1 mAb susceptibility.** (A) Antibody titration curves summarizing complement-dependent killing of HB19-R1 viability reporter strains expressing OspA variants with polymorphisms at position 107. Complement-dependent killing assays were performed as described in the materials and methods section using anti-OspA Bin1 mAbs and *B. burgdorferi* HB19-R1 viability reporter strains expressing "WT" OspA variants from *B. burgdorferi B31* (ST1), *B. valaisiana* VS116 (IST15), and *B. turdi* TPT2017 (IST16), or mutated derivatives containing deletions, insertions, or substitutions at residue 107. (B) Susceptibility of HB19-R1 reporter strains harboring OspA variants with polymorphisms at position 107 to complement dependent killing mediated by 66.7 nM of Class II & Class III Bin 1 mAbs. All data shown includes at least three independent experiments per strain with data normalized as described.
(PDF)

**S10 Fig. OspA β-Strand Labels, and Combined Epitope of Bin 1 mAbs.** Ribbon diagram of OspA depicting close up of the CBS with residues conserved within the epitopes of 857-2, 221-5, 221-11, and 227-1 labeled and shaded in blue. Strand numbers are labeled in red.
(PDF)

**S11 Fig. Human and Guinea Pig-derived complement can be used interchangeably in SBAs with *B. burgdorferi* HB19-R1 reporter strains.** Complement-dependent bactericidal assays were performed using HB19-R1 reporter strains and mAb 857-2 with either 2.5% guinea pig complement or 5% human complement (Pel-Freeze), under conditions described in the materials and methods section. Data shown encompasses 3–5 independent experiments. Differences in strain susceptibility to mAb 857-2 between complement sources were assessed across all strains by two-way ANOVA followed by Sidak's multiple-comparisons test. No significant differences in susceptibility were observed.
(PDF)

**S1 File. Files associated with OspA phylogenetic tree construction.** This dataset includes the multiple sequence alignment, phylogenetic tree, and phylogenetic tree formatting files used to construct the OspA phylogenetic tree shown in Fig 6.
(7Z)

**S2 File. Vector map and sequence of pGW217.** This dataset includes the vector map and annotated sequence file for the OspAST1 viability reporter plasmid used to construct all other OspA-expressing mscarlet-I reporter plasmids described in this study.
(ZIP)

## Acknowledgments

We gratefully acknowledge Dr. John Leong (Tufts University) for providing *Borrelia* isolates and Dr. Meredith Finn (Moderna) for providing recombinant OspA proteins. We thank the Wadsworth Center's Cell culture and media core for preparation of BSK II media, the Genomics core for DNA sequencing, and Drs. Renjie Song and Jennifer Yates of the Immunology Core facility for assistance in optimizing flow cytometry parameters. X-ray analysis as conducted at the Northeastern Collaborative Access Team beamlines was funded by the NIH's National Institute of General Medical Sciences (P30 GM124165 to MJR) and the Eiger 16M detector on the 24-ID-E beam line was funded by a NIH Office of Research Infrastructure Programs (ORIP) High-End Instrumentation (HEI) grant (S10OD021527 to MG).

## Author contributions

**Conceptualization:** Graham G Willsey, Michael J Rudolph, Lisa A Cavacini, David J Vance, Nicholas J. Mantis.

**Data curation:** Graham G Willsey, Michael J Rudolph.

**Formal analysis:** Graham G Willsey, Michael J Rudolph, David J Vance, Nicholas J. Mantis.

**Funding acquisition:** Michael J Rudolph, Nicholas J. Mantis.

**Investigation:** Graham G Willsey, Michael J Rudolph, Carol Lyn Piazza, Yang Chen, Grace Freeman-Gallant, Lisa A Cavacini, David J Vance, Nicholas J. Mantis.

**Methodology:** Graham G Willsey, Michael J Rudolph, Carol Lyn Piazza, Yang Chen, Grace Freeman-Gallant, Lisa A Cavacini, David J Vance.

**Project administration:** Nicholas J. Mantis.

**Resources:** Graham G Willsey, Carol Lyn Piazza, Lisa A Cavacini.

**Supervision:** Michael J Rudolph, Lisa A Cavacini, Nicholas J. Mantis.

**Validation:** Michael J Rudolph, Lisa A Cavacini, David J Vance.

**Visualization:** Michael J Rudolph.

**Writing – original draft:** Graham G Willsey, Michael J Rudolph, Carol Lyn Piazza, Lisa A Cavacini, David J Vance, Nicholas J. Mantis.

**Writing – review & editing:** Graham G Willsey, Michael J Rudolph, Nicholas J. Mantis.

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
