## [Decision Letter · Decision Letter 0]

22 Dec 2025

PPATHOGENS-D-25-02904

A Broadly Conserved Protective Epitope on the Lyme Disease Vaccine Antigen, OspA

PLOS Pathogens

Dear Dr. Mantis,

Thank you for submitting your manuscript to PLOS Pathogens. Three reviewers and two members of the editorial board agree that this is a high-quality, innovative, and timely study addressing a major unmet need in Lyme disease vaccine development. While the overall dataset is compelling, concerns were raised regarding the reliance on engineered reporter strains and in vitro assays for several core conclusions. In particular, additional evidence demonstrating epitope accessibility in tick-relevant environments was noted as an important element to strengthen the transmission-blocking claims.

In addition, Pfeifle et al. just published a study in NPJ Vaccines describing a polyvalent mRNA vaccine targeting OspC. It may be of interest to discuss this work in the manuscript.

Finally, the name *Borreliella* is not universally accepted within the community. I would therefore strongly suggest using is not universally accepted within the community. I would therefore strongly suggest using *Borrelia* throughout the manuscript, or at least explicitly indicating both names at first mention in the main text and the abstract.throughout the manuscript, or at least explicitly indicating both names at first mention in the main text and the abstract.

After careful consideration, we feel that it has merit but does not fully meet PLOS Pathogens's publication criteria as it currently stands. Therefore, we invite you to submit a revised version of the manuscript that addresses the points raised during the review process.

We look forward to receiving your revised manuscript.

Kind regards,

Sébastien Bontemps-Gallo

Academic Editor

PLOS Pathogens

Thomas Guillard

Section Editor

PLOS Pathogens

Sumita Bhaduri-McIntosh

Editor-in-Chief

PLOS Pathogens

orcid.org/0000-0003-2946-9497

Michael Malim

Editor-in-Chief

PLOS Pathogens

orcid.org/0000-0002-7699-2064

**Journal Requirements:**

At this stage, the following Authors/Authors require contributions: Graham G Willsey, Michael Rudolph, Carol Lyn Piazza, Yang Chen, Grace Freeman-Gallant, Lisa A Cavacini, David J Vance, and Nicholas J Mantis. Please ensure that the full contributions of each author are acknowledged in the "Add/Edit/Remove Authors" section of our submission form.

https://journals.plos.org/plospathogens/s/submission-guidelines#loc-parts-of-a-submission

5) We have noticed that you have uploaded Supporting Information files, but you have not included a list of legends. Please add a full list of legends for your Supporting Information files after the references list.

**Reviewers' Comments:**

Reviewer's Responses to Questions

**Part I - Summary**

Reviewer #1: More than three decades after the demonstration that immunization with outer surface protein A (OspA) elicits protective antibodies, there is still no commercially available Lyme disease vaccine. Current OspA vaccine candidates undergoing clinical trials use polyvalent recombinant constructs fusing protective regions of OspAs derived from different serotypes. This meticulously executed, innovative, well-presented study uses a powerful combination of genetically engineered OspA reporter strains, borreliacidal human monoclonal antibodies, and structural analysis of OspA-Fab complexes to define a broadly conserved region of OspA’s central beta sheet that could become a universal Lyme disease vaccine. Some concerns noted elsewhere need the attention of the authors.

Reviewer #2: This manuscript presents a robust and comprehensive study identifying a broadly conserved protective epitope within the central β-sheet of OspA, a major antigenic target for Lyme disease vaccines. The authors integrate genetic engineering, functional immunology, and structural biology in an elegant manner. The creation of an isogenic panel of Borrelia burgdorferi reporter strains expressing OspA serotypes ST1–7 represents a methodological strength, enabling controlled and standardized comparison of cross-serotype antibody binding and complement-dependent borreliacidal activity. The structural analyses of Fab–OspA complexes are of high quality and provide strong mechanistic insight into epitope recognition.

This manuscript presents an ambitious and technically sophisticated study aiming to define a broadly conserved protective epitope within the central β-sheet of OspA, a major antigen for Lyme disease vaccine development. The integration of functional assays, structural biology, and engineered Borrelia reporter strains is a notable strength, and the identification of a potentially cross-serotype epitope is highly relevant to current efforts in next-generation OspA-based vaccine design.

Despite these strengths, several core conclusions rely heavily on engineered reporter strains and in vitro assays. The study would substantially benefit from data that demonstrate epitope accessibility and antibody activity in tick-relevant environments, as well as validation of the proposed susceptibility determinant Lys107 in native Borrelia isolates rather than only in a heterologous expression system. Additionally, variability in OspA expression across constructs has not been quantified, which raises concerns about whether functional differences reflect sequence-specific effects or expression levels.

The work is impactful and of high potential significance, but several experimental gaps currently limit the strength and generalizability of the conclusions. Addressing these issues is essential before acceptance.

Reviewer #3: This is an excellent study showing that the central core of B. burgdorferi OspA (in the vicinity of B-sheets 6-10, aa82-146) is conserved across all major OspA serotypes. In addition, they identify the invariant aa residues that contribute to antibody recognition (Glu100, Asp105 and Lys107). The results are transformational for development of cross-reactive OspA based LD vaccines. Although the first OspA vaccines were based in B. burgdorferi B31 OspA (ST1), which is fully protective against all Bb strains transmitted by vector Ixodes in the United States (hence a hint of cross-reactive protection, and older evidence identifies OspA’ central B-sheet as cross-reactive), current OspA based LD vaccine iterations contain 7 serotypes to include the European Borreliella genospecies. Several companies are pursuing development of multivalent OspA vaccines, and the most advanced is Valneva/Pfizer recombinant protein which is comprised of 7 C-terminus fractions of OspAs ST1 to ST7 and is undergoing Phase III clinical trials. The study is very well designed, well executed, and the reporting of the results is rigorous. The scientific community working on OspA based vaccines for LD will be grateful for the set of B. burgdorferi spirochetes expressing each of the 7 OspA seroptypes as a tool to test vaccine efficacy in vitro by borreliacidal assay.

**Part II – Major Issues: Key Experiments Required for Acceptance**

Please use this section to detail the key new experiments or modifications of existing experiments that should be absolutely required to validate study conclusions.required to validate study conclusions.

Reviewer #1: None needed.

Reviewer #2: While the manuscript is strong and presents compelling data, several essential experimental gaps must be addressed before the conclusions can be fully supported. These additional experiments are necessary for acceptance, as they directly impact the central claims of epitope accessibility, cross-serotype generalizability, and mechanistic interpretation.

1. Functional confirmation of epitope accessibility in a tick-derived context

The study demonstrates broad mAb activity using in vitro reporter strains, but OspA expression and conformation differ substantially in the tick midgut. Since the proposed epitope is relevant for vaccine-mediated transmission blocking, the authors should provide at least one line of experimental evidence supporting epitope accessibility in a tick-relevant environment.

Required experiment:

Evaluate binding of one representative Class I mAb (e.g., 857-2 or 221-20) to Borrelia in dissected tick midguts or to in vitro tick cell–associated OspA, using flow cytometry or immunofluorescence.

Rationale: Validates that the structurally defined epitope remains exposed during natural transmission.

2. Validation of Lys107’s functional role in a native OspA background

The importance of Lys107 is well supported using engineered OspA variants in the HB19-R1 background. However, the manuscript extrapolates this to natural genospecies without direct functional validation.

Required experiment:

Complement-dependent killing assays using native isolates of at least one Lys107-deficient species (e.g., B. lusitaniae or B. turdi) and the corresponding engineered “Lys107-restored” mutant.

Rationale: Demonstrates that Lys107 determines susceptibility in authentic clinical/genomic backgrounds, not only in the reporter system.

Statistical treatment of EC₅₀ differences across serotypes

The heatmaps and EC₅₀ values are compelling, but the manuscript lacks statistical comparison across serotypes and mAbs.

Required modification of existing data:

Provide statistical analysis (e.g., ANOVA or pairwise comparisons) for EC₅₀ differences across OspA serotypes within each mAb class.

Rationale: Strengthens claims regarding potency hierarchy and cross-serotype differences.

Reviewer #3: No major issues

**Part III – Minor Issues: Editorial and Data Presentation Modifications**

Reviewer #1: 1. The authors vacillate between describing the mab binding region as an epitope or epitopes (e.g., line 197 "overlapping but distinct epitopes"). Per the title, the data give the general impression that the mAbs recognize one epitope with variable binding due to sporadic amino acid substitutions in OspA and Fab variable chains. To help clarify this issue and place the results in a broader context, it would be interesting to know about B cell epitope predictions for this central region of OspA and how they are affected by the amino acid substitutions that impact mAB reactivity.

2. Abstract, line 24. "Debilitating" is a bit excessive as a general descriptor for Lyme disease. The vast majority of patients with Lyme disease do not have debilitating manifestations.

3. Introduction, line 60. Substitute "may result in" for "may involve".

4. Introduction, lines 92-93. Are the authors sure about this? The abstract to reference 17 states that protection was not observed when mice were immunized with a heterologous OspA.

5. Results, line 124-125. This sentence could be worded better to more precisely state what about the strains is being validated.

6. Results, line 135 and elsewhere. What does "Bin 1" mean? I have not been able to find a definition of the term.

7. Results, lines 147-163. The data clearly indicate that OspA expression levels are the same in both strains but this should be clearly stated given the potential impact of this variable on the results.

8. Figure 3. It would be helpful to indicate mAb class in the figure and label the reactive loops.

9. Figure 6B. It would be helpful if the authors could modify the figure to make the strains with the Lys 107 polymorphisms easier to spot.

10. Discussion, lines 374-375. Suggest adding this information to the appropriate place in the Results since it is an obvious question raised by the data presented.

Reviewer #2: Writing and clarity

Some figure legends (particularly Figures 4–7) would benefit from additional detail and simplification.

Supplementary figure captions (e.g., S7, S10) should better explain what the reader is expected to interpret.

Methods

The choice of guinea pig complement should be justified more explicitly given known strain-specific complement sensitivities.

Please clarify whether recombinant OspA variants used in binding assays retain lipidation or are unlipidated forms.

Data availability

Structural PDB accession numbers should be explicitly referenced in the Data Availability Statement, not only in Methods.

Figures

Figure 1: consider adding statistical information to binding differences.

Figure 2: adding a summary EC₅₀ table in the main text or supplement would improve readability.

Figure 6: the phylogenetic tree is difficult to read at current scale; consider including zoomed panels.

Reviewer #3: Discussion: It may be useful to add a few words in the discussion on the residues that comprise B-sheets 6-10 because I found these buried in the methods section and it is an important fact to highlight vis a vis Valneva/Pfizer C-terminus vax currently in Phase III clinical trials. I am pretty confident these B-sheets fall outside the C-terminus of OspA for the most part. This will open-up discussion and fine-tuning of OspA vaccines, old and new. If I am wrong then that question is resolved with one statement.

PLOS authors have the option to publish the peer review history of their article (what does this mean?). If published, this will include your full peer review and any attached files.). If published, this will include your full peer review and any attached files.

.

Reviewer #1: No

Reviewer #2: **Yes:** Dr CHOUMET ValérieDr CHOUMET Valérie

Reviewer #3: **Yes:** Maria Gomes-SoleckiMaria Gomes-Solecki

**Figure resubmission:**
---

## [Decision Letter · Decision Letter 1]

24 Mar 2026

PPATHOGENS-D-25-02904R1

A Broadly Conserved Protective Epitope on the Lyme Disease Vaccine Antigen, OspA

PLOS Pathogens

Dear Dr. Mantis,

Thank you for submitting your manuscript to PLOS Pathogens.

It may be helpful to revise the title for consistency with the revised text, which suggests multiple overlapping protective epitopes; for example, removing the article “a” could help avoid implying a single epitope.

Therefore, we invite you to submit a revised version of the manuscript that addresses the points raised during the review process.

We look forward to receiving your revised manuscript.

Kind regards,

Sébastien Bontemps-Gallo

Academic Editor

PLOS Pathogens

Thomas Guillard

Section Editor

PLOS Pathogens

Sumita Bhaduri-McIntosh

Editor-in-Chief

PLOS Pathogens

orcid.org/0000-0003-2946-9497

Michael Malim

Editor-in-Chief

PLOS Pathogens

orcid.org/0000-0002-7699-2064

**Journal Requirements:**

**Reviewers' Comments:**

Reviewer's Responses to Questions

**Part I - Summary**

Reviewer #1: More than three decades after the demonstration that immunization with outer surface protein A (OspA) elicits protective antibodies, there is still no commercially available Lyme disease vaccine. Current OspA vaccine candidates undergoing clinical trials use polyvalent recombinant constructs fusing protective regions of OspAs derived from different serotypes. This meticulously executed, innovative, well-presented study uses a powerful combination of genetically engineered OspA reporter strains, borreliacidal human monoclonal antibodies, and structural analysis of OspA-Fab complexes to define a broadly conserved region of OspA’s central beta sheet that could become a universal Lyme disease vaccine.

Reviewer #2: The authors have carefully addressed the concerns raised during the first round of review. The revisions have significantly improved the clarity of the manuscript as well as the presentation of the data. The additional explanations and modifications provided by the authors satisfactorily address the main points that were previously raised.

Overall, the study presents solid experimental work and provides valuable insights into the topic investigated. The manuscript is well structured, the methodology is appropriate, and the conclusions are supported by the data presented. In its revised form, the manuscript is suitable for publication.

Reviewer #3: (No Response)

**Part II – Major Issues: Key Experiments Required for Acceptance**

Please use this section to detail the key new experiments or modifications of existing experiments that should be absolutely required to validate study conclusions. required to validate study conclusions.

Generally, there should be no more than 3 such required experiments or major modifications for a 'Major Revision' recommendation. If more than 3 experiments are necessary to validate the study conclusions, then you are encouraged to recommend 'Reject'.

Reviewer #1: None needed

Reviewer #2: The authors have satisfactorily addressed the major concerns raised in the initial review. The clarifications and revisions provided in the revised manuscript adequately support the conclusions of the study.

At this stage, no additional major experiments appear necessary, and I recommend acceptance of the manuscript.

Reviewer #3: (No Response)

**Part III – Minor Issues: Editorial and Data Presentation Modifications**

Reviewer #1: In response to this Reviewer's Comment 1.1 about whether the mabs recognize multiple epitopes, the authors stated 'The reviewer makes an excellent point and have revisited the text and corrected wording to make the point that there are likely

multiple overlapping protective epitopes within this region of OspA.' However, the title of the revised manuscript 'A Broadly Conserved Protective Epitope on the Lyme Disease Vaccine Antigen, OspA' has not been amended to reflect this textual revision.

Reviewer #2: Only a few minor editorial adjustments may still improve the clarity of the manuscript. These include minor language polishing and ensuring consistency in figure legends and formatting throughout the manuscript. These points can easily be addressed during the final editorial revision.

Reviewer #3: (No Response)

PLOS authors have the option to publish the peer review history of their article (what does this mean?). If published, this will include your full peer review and any attached files.). If published, this will include your full peer review and any attached files.

**Do you want your identity to be public for this peer review?** For information about this choice, including consent withdrawal, please see our  For information about this choice, including consent withdrawal, please see our Privacy Policy..

Reviewer #1: **Yes:** Justin RadolfJustin Radolf

Reviewer #2: **Yes:** Dr Valérie CHOUMETDr Valérie CHOUMET

Reviewer #3: **Yes:** Maria Gomes-SoleckiMaria Gomes-Solecki

**Figure resubmission:**

After uploading your figures to PLOS’s NAAS tool - https://ngplosjournals.pagemajik.ai/artanalysis, NAAS will process the files provided and display the results in the 'Uploaded Files' section of the page as the processing is complete. If the uploaded figures meet our requirements (or NAAS is able to fix the files to meet our requirements), the figure will be marked as 'fixed' above. If NAAS is unable to fix the files, a red 'failed' label will appear above. When NAAS has confirmed that the figure files meet our requirements, please download the file via the download option, and include these NAAS processed figure files when submitting your revised manuscript.
---

## [Editor Report · Decision Letter 2]

7 Apr 2026

Dear Dr. Mantis,

We are pleased to inform you that your manuscript 'Broadly Conserved Protective Epitopes on the Lyme Disease Vaccine Antigen, OspA' has been provisionally accepted for publication in PLOS Pathogens.

Best regards,

Sébastien Bontemps-Gallo

Academic Editor

PLOS Pathogens

Thomas Guillard

Section Editor

PLOS Pathogens

Sumita Bhaduri-McIntosh

Editor-in-Chief

PLOS Pathogens

orcid.org/0000-0003-2946-9497

Michael Malim

Editor-in-Chief

PLOS Pathogens

orcid.org/0000-0002-7699-2064
---

## [Editor Report · Acceptance letter]

Dear Dr. Mantis,

We are delighted to inform you that your manuscript, "Broadly Conserved Protective Epitopes on the Lyme Disease Vaccine Antigen, OspA," has been formally accepted for publication in PLOS Pathogens.

Best regards,

Sumita Bhaduri-McIntosh

Editor-in-Chief

PLOS Pathogens

orcid.org/0000-0003-2946-9497

Michael Malim

Editor-in-Chief

PLOS Pathogens

orcid.org/0000-0002-7699-2064